# UAV survey method to monitor and analyze geological hazards: The Case study of the mud volcano of Villaggio Santa Barbara, Caltanissetta (Sicily)

Fabio Brighenti[1], Francesco Carnemolla [1], Danilo Messina[3] & Giorgio De Guidi[1][2]

[1] Department of Biology, geology and Environmental Sciences, University of Catania, Catania, 95129, Corso Italia 55 – 57, Italy

[2] CRUST-Interuniversity Center for 3D Seismotectonics with territorial applications - UR-UniCT, Catania, 95129, Corso Italia 55 – 57, Italy

[3] Indipendent Researcher

*Correspondence to:* Giorgio De Guidi (deguidi@unict.it)

**Abstract.** All active geological processes generally determine effects on the ground surfaces such as uplift, subsidence and shear lineaments. Nowadays remote sensing represents a key tool for the evaluation and monitoring of natural hazards. The use of Unmanned Aerial Vehicles (UAVs) in relation to observation of natural hazards, encompasses three main stages: pre-post event data acquisition, monitoring and risk assessment. The mud volcano of Santa Barbara (Municipality of

Caltanissetta, Italy), represents a dangerous site because on 11th August 2008 a paroxysmal event caused serious damage to infrastructures within a range of about 2 km. The main precursors to mud volcano paroxysmal events are uplift and the development of structural features with dimensions ranging from centimetres to decimetres. Here we present a methodology for monitoring deformation processes that may be precursory to paroxysmal events at the Santa Barbara mud volcano. This methodology is based on i) the data collection, ii) the Structure from Motion (SfM) processing chain and iii) the M3C2-PM

algorithm for the comparison between point clouds and uncertainty analysis with a statistical approach. The objective of this methodology is to detect precursory activity by monitoring deformation processes with centimetre scale precision and a temporal frequency of 1 - 2 months.

## 1. Introduction

In recent decades, both high-resolution digital photographs and Structure from Motion (SfM) softwares has allowed the

generation of high-quality topographic information. In geosciences many studies have been dedicated to morphological processes (Castillo et al., 2012; James and Robson, 2012; James and Varley, 2012; Amici et al. 2013b; Casella et al., 2014; Gomez-Gutierrez et al., 2014; James and Robson, 2014b; Lucieer et al., 2014; Ryan et al., 2015; Westoby et al., 2015; Woodget et al., 2015; Eltner et al., 2015; Dietrich, 2016; Smith et al., 2016; Javernick et al., 2016; Walter et al., 2018b; Deng et al. 2019). Applications include runoff laboratory trials (Morgan et al., 2017), applied geology (Niethammer et al., 2012;

Russell, 2016; Saito et al., 2018), geomorphology (Javernick et al., 2014; Snapir et al., 2014; Dietrich, 2015; Smith and

Vericat, 2015; Bakker and Lane, 2016; Dietrich, 2016 a/b; Mercer and Westbrook, 2016; Pearson et al., 2017; Prosdocimi et al., 2017; Marteau et al., 2016; Balaguer-Puig et al., 2017; Vinci et al., 2017; Heindel et al., 2018; Seitz et al., 2018), glaciology (Immerzeel et al., 2017; Piermattei et al., 2016), coastal morphology (James and Robson, 2012; Casella et al., 2016; Brunier et al., 2016), volcanology (James and Robson, 2012; Bretar et al., 2013; Müller et al. 2017; Giordan et al.,
2017, 2018; Carr et al., 2018; Favalli et al., 2018; Witt et al. 2018; Andaru and Rau, 2019; Bonali et al. 2019; De Beni et al. 2019) and geophysics (Amici et al., 2013a; Greco et al., 2016; Di Felice et al., 2018; Zahorec et al., 2018; Federico et al., 2019). SfM is commonly used in the cultural heritage field for 3D reconstruction (Sapirstein, 2016, 2018; Sapirstein and Murray, 2017; Jalandoni et al., 2018). The monitoring of active geological processes is a preventive action in risk mitigation (Stöcker et al., 2017; Turner et al., 2017b; Diefenbach et al., 2018; Rosa et al., 2018; Deng et al. 2019). Disasters occur when
two factors - hazard and vulnerability - coincide. The risk is proportional to the magnitude of the hazards and the vulnerability of the affected population. Among the deformation monitoring systems, the photogrammetry technique from Unmanned Aerial Vehicles (UAVs) is becoming more widely used thanks to the high efficiency in data acquisition, the low cost compared to traditional techniques and the acquisition of high resolution images (Harwin and Lucieer, 2012; James and Robson, 2012; Westoby et al., 2012; Fonstad et al., 2013; Javernick et al., 2013; Johnson et al., 2014; James et al., 2017/a/b;
James et al., 2020). This technique is important to study catastrophic natural events such as floods, earthquakes, landslides, etc. Different acquisition methods and the ability to obtain high spatial (centimetre) and temporal resolution (hours or days) (Boccardo et al. 2015) enable the acquisition of detailed information on the evolution of the landscape, therefore UAVs are an effective and complementary tool for field investigations. Furthermore, UAVs have other advantages including: (i) the ability to fly at low altitudes, (ii) the ability to reach remote locations, (iii) the ability to host multiple sensors (cameras,
Lidar, thermal imaging cameras, navigation / inertial sensors, etc.), (iv) the ability to capture images at different angles, and (v) the flexibility to carry out monitoring operations on a small, medium and large scale (Jordan et al. 2017). Ground Control Points (GCPs) are used to improve the accuracy of the resulting data. Therefore, recognisable points on the UAV imagery are measured with a high-precision surveying device to georeference the data. In this process a correct number of GCPs is required which lead to a greater accuracy of the resulting data (point clouds, 3D grid, orthomosaic or Digital Surface Model
(DSM)). The precision of the resulting data is also controlled by other variables, such as: the focal distance of the camera; flight path and flight altitude; the orientation of the camera; the picture quality; the processing chain and the category of UAV system (fixed or rotary wings).

In this paper, we present the results and analysis of the surface deformation monitoring of the mud volcano of Santa Barbara (Caltanissetta, central Sicily) (Fig.1). We have applied the statistical analysis of significant changes with Level of Detection
$_{95\%}$ (LoD $_{95\%}$). In detail, we used precision maps and the M3C2-PM (Lague et al., 2013; James et al., 2017b) algorithm to determine the surface variations. The statistical analysis allows to verify i) the uncertainty between the different surveys, ii) the spatial variability of the accuracy in the surveys (James et al., 2017/b), iii) the quality of the georeferencing of the surveys based on the number of GCPs.

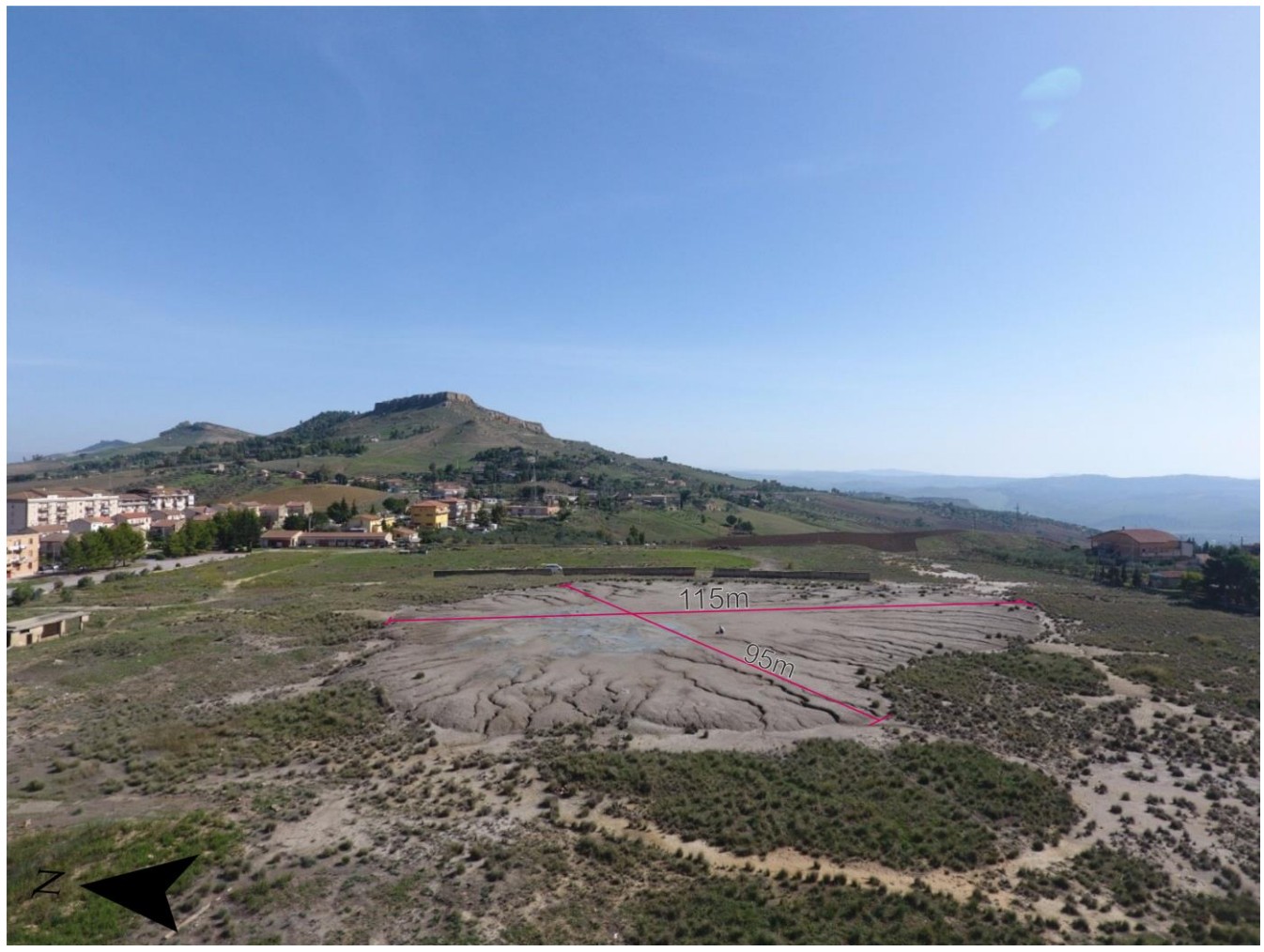

**Figure 1: Aerial view of the Santa Barbara mud volcano. Photo taken by the UAV from the west side of the mud volcano.**

The mud volcano of Santa Barbara is located within the Caltanissetta foredeep basin of the Apennines-Maghrebian collisional chain which developed from the Late Miocene to the Quaternary, along the border of the converging Eurasia-
Nubia plate (Catalano et al., 2008; Dewey et al., 1989; Serpelloni et al., 2007). This structural domain is formed by a foreland fold and thrust belt involving the deposition of clastic sediments which were gradually deformed from the late Miocene to the Pleistocene (Monaco and Tortorici, 1996; Lickorish et al., 1999, and references therein).

According to Madonia et al., (2011) mud volcanoes are most of the time in stasis, but they represent a preferential way for rising fluids rich in methane and sludge, therefore they can be considered a risk to urbanized areas or sites with an economy dedicated to natural attractions.

At several mud volcanoes (e.g. Ayaz-Akhtarma and Khara Zira Island mud volcanoes in Indonesia), certain geomorphic/structural features have been observed within the year preceding a paroxysmal event (Antonelli et al., 2014; Madonia et al., 2011). Also the area of the Santa Barbara volcano was affected in 2008 by paroxysmal mud eruption which was preceded by deformation features (Fig.2). Moreover, the surface of the mud volcanic cone is incised by a drainage system (Fig.1) characterised by hydrographic basins with elongated dendritic geometry arranged to a centrifugal development from the areas of the summit craters towards the lower slopes of the volcano complex. The higher order of the thalwegs present deep recessed meanders (landscape rejuvenation process). This morphometric structure is typical of uplifting areas and therefore relative decrease of the base level. This suggests an inflection process of the volcano ground surface induced by increase of fluid pressure inside of shallower stagnation chamber which is located at a depth of about 30 m (Imposa et al., 2018). The stagnation chamber has a "sill-like" geometry, a radius of about 50 m and a thickness of about 30 m. This morphostructural configuration supported by geophysical data configures the active geological structure as a high potential geological hazard.

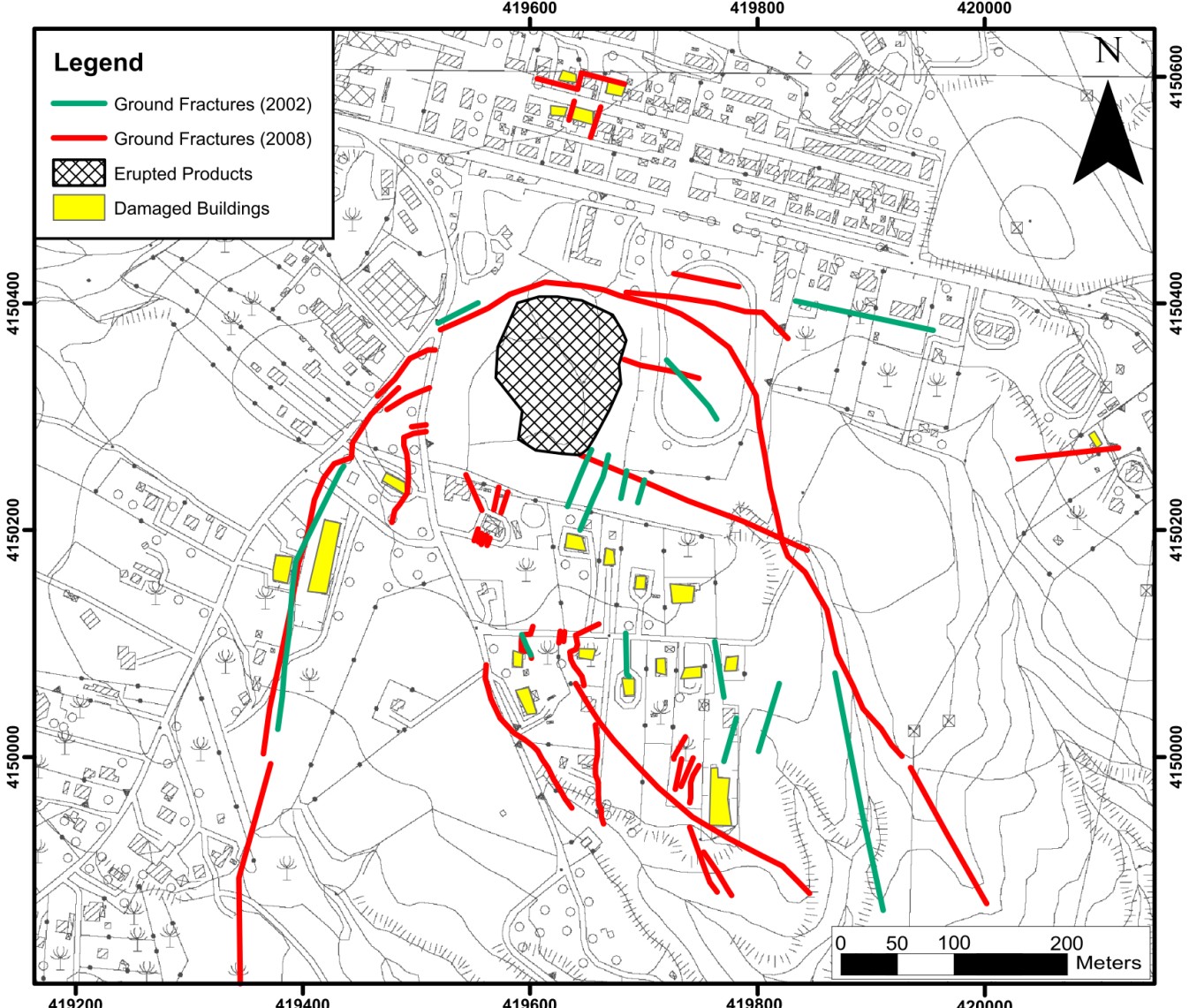

**Figure 2: Cartographic extract with the location of the major fractures (in red) detected on the ground and the damaged structures (in yellow) related to the paroxysmal event of 2008. In green fractures detected in 2002. The base map is Carta Tecnica Regionale 2008 (CTR). The reference system is WGS 84 / UTM zone 33N.**

On the surface, fractures and shear lineaments extend outside the erupted mud area, (Madonia et al., 2011; Bonini et al. 2012; INGV, 2008a; Regione Siciliana, 2008) and they highlight the high stress and strain environment induced by the mud volcano (Fig.2). Such structures have been detected in 2002 and 2008 we speculate that they are still active (Fig.2). This development has often been a precursor of paroxysmal events such as the 11th August 2008 event (INGV, 2008a; Regione Siciliana, 2008).

## 2. Methods


### 2.1 Local Network

In order to monitor active deformation in the mud volcano area, a local GNSS network was created according to the criteria described by De Guidi et al (2017), in particular ensuring i) the basic requirement of spatial and temporal stability ii) absence

of possible gravitational instabilities in both static and dynamic conditions at sites and iii) a panoramic and elevated position for the Theodolite Total Station (TST).

According to these criteria two GNSS benchmarks were created: CTN0 and CTN1, located on the roof of a building on the northern sector of the studied area (Fig.3).

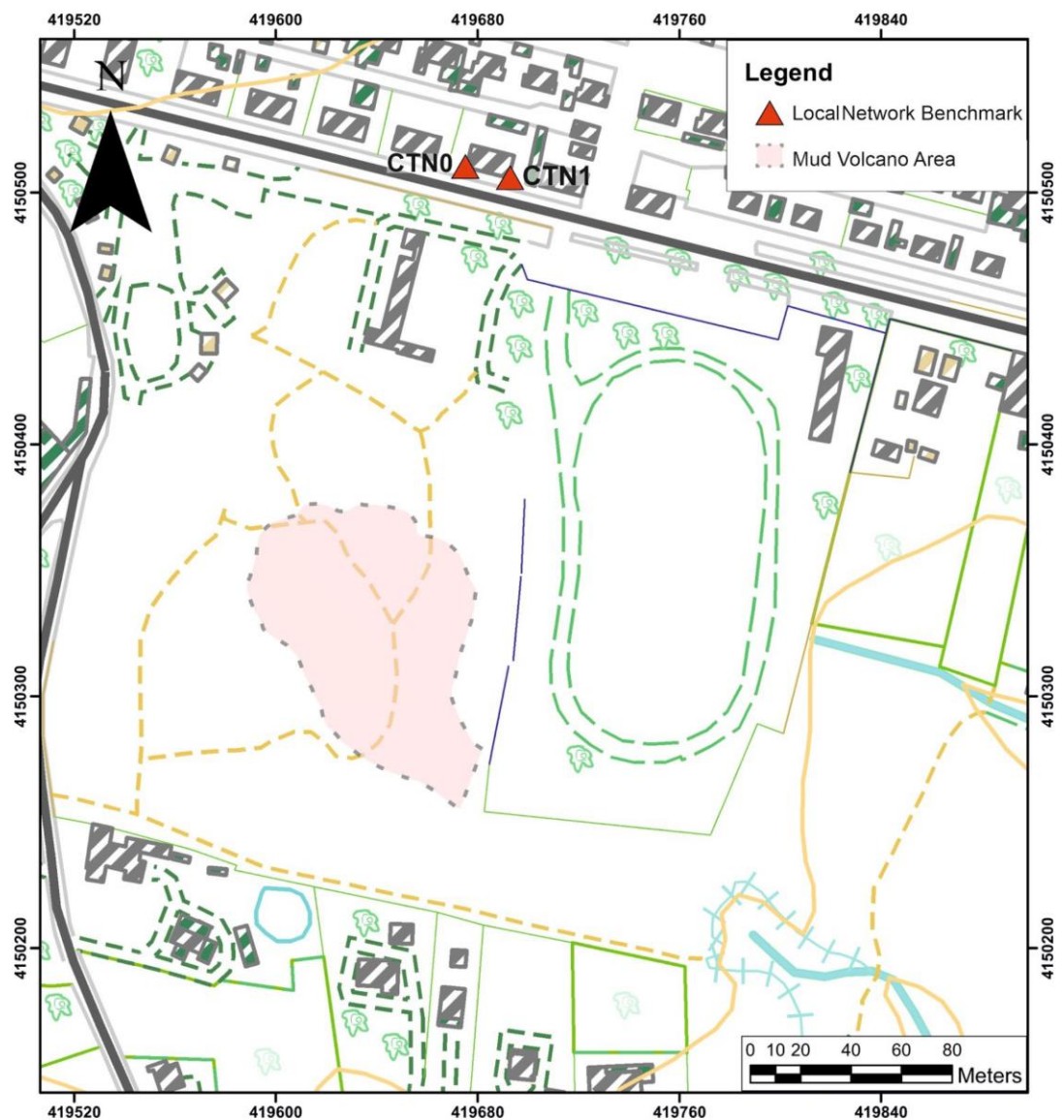


**Figure 3: Chorography of the eastern periphery of the inhabited area of Caltanissetta (Santa Barbara Village). The base map is Carta Tecnica Regionale 2008 (CTR). The reference system is WGS 84 / UTM zone 33N.**

The benchmarks were surveyed using double frequency (L1/L2) receivers (TOPCON Hiper V and HiPer SR) in static mode.
Once the stability of the benchmarks has been assessed, we surveyed CTN0 and CTN1 5 and 2 times respectively.

Post processing of GNSS data was carried out by AUSPOS Online service (Geoscience Australia, 2011; Jia, M et al., 2014).

To process the CTN0 data of 2018/02/28 survey, AUSPOS used 15 IGS stations to compute the baselines: ANKR, BOR1, BRUX, BUCU, GANP, GRAS, GRAZ, LROC, MAT1, MEDI, SOFI, TLSE, VILL, YEBE and ZIM2 with an average of ambiguity resolution of 90.0%, position uncertainty (95% C.L.) are respectively of 0.005 m, 0.005 m, 0.016 m for East, North and Ellipsoidal height.

To process the CTN1 data of 2018/11/26 survey, AUSPOS used 15 IGS stations to compute the baselines: ANKR, BOR1, BRUX, BUCU, GANP, GRAS, GRAZ, LROC, MAT1, MEDI, SOFI, TLSE, VILL, YEBE and ZIM2 with an average of ambiguity resolution of 84.5%, position uncertainty (95% C.L.) are respectively of 0.007 m, 0.007 m, 0.024 m for East, North and Ellipsoidal height.

Finally, the optimal ITRF2014-UTM33N coordinates have been definitively assigned to CTN0 and CTN1.

## 2.2 Ground Control Points (GCPs)

Various acquisition methods of Ground Control Points (GCPs) were tested in order to define the most suitable one. The main function of GCPs is to geo-reference outcomes from SfM.

Initially (2016-2018) we used only GNSS receivers in different configurations: Real Time Kinematics (RTK) and Static Ultra Rapid. The GCPs were made of 50 cm x 50 cm alveolar polypropylene square targets. Using these configurations (Tab. 1), errors ranging from centimetre to decimetre were recorded. In these early phases, errors were only computed by SfM software PhotoScan (v 1.4.5.7554).

PhotoScan provides different types of error estimation: XY error (m) - root mean square error for horizontal coordinates for a GCP location; Z error (m) - error for elevation coordinate for a GCP location; Error X, Y and Z (m) - root mean square error for X, Y, Z coordinates for a GCP location; Error Img (pix) - root mean square error for X, Y coordinates on an image for a GCP location averaged over all the images; Total Error (m) - implies averaging over all the GCP locations.

Using Static Ultra Rapid mode, the total error was about 18 cm, whereas using the RTK configuration the total error were reduced to approximately 4 cm. (Brighenti et al, 2018).

From 2018 the use of the TopCon DS-103 TST was introduced, obtaining lower values on the GCP errors than those measured with the GNSS technique (Tab.1). With this measurement technique the total error has been reduced to about 3 cm. On the first 3 campaigns, we used 6 GCPs to georeferenced the cloud points (Fig.4). In the following section, these first 3 campaigns will be not considered in the computation of the significant changes for their incomparable to the last campaigns. Since 2019, we preferred to use only the TST for the GCP survey and, according to Tahar et al. (2013), the

145    number of GCPs has been increased (Tab.1). Considering these two improvement, the total error has been reduced to about 1.4 cm and in the last campaign about 0.7 cm.

| DATE | METHOD OF SURVEY | GCP NUMBER | X ERROR (CM) | Y ERROR (CM) | Z ERROR (CM) | TOTAL ERROR (CM) | IMG ERROR (PIX) |
|---|---|---|---|---|---|---|---|
| 2018/02/28 | Static Ultra Rapid | 6 | 12.06 | 8.85 | 10.02 | 18.01 | 0.587 |
| 2018/04/16 | RTK | 6 | 2.18 | 1.79 | 3.34 | 4.29 | 0.326 |
| 2018/04/16 | TST | 6 | 2.04 | 1.39 | 1.68 | 2.99 | 0.325 |
| 2019/07/29 | TST | 29 | 0.78 | 0.95 | 0.72 | 1.43 | 0.244 |
| 2019/09/13 | TST | 30 | 0.79 | 0.84 | 0.86 | 1.45 | 0.316 |
| 2019/10/14 | TST | 31 | 0.78 | 0.66 | 0.78 | 1.29 | 0.311 |
| 2020/01/13 | TST | 31 | 0.95 | 0.82 | 0.77 | 1.48 | 0.237 |
| 2020/06/15 | TST | 26 | 0.43 | 0.39 | 0.45 | 0.73 | 0.249 |

**Table 1: Average RMSE of GCPs obtained for each survey and for each technique used to determinate the GCPs coordinates: X (Easting), Y (Northing), Z (Altitude) and the Total Error. The image residual (IMG ERROR, in the table) is shown.**

150

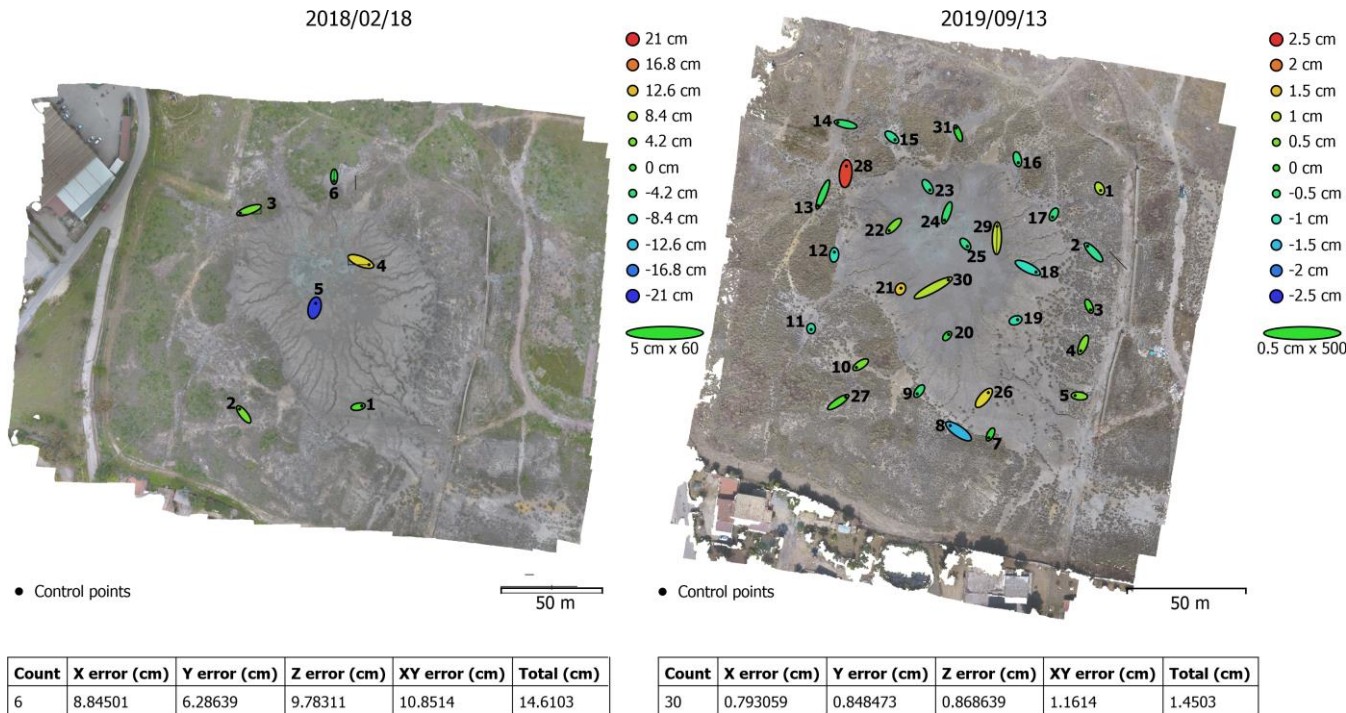

| Count | X error (cm) | Y error (cm) | Z error (cm) | XY error (cm) | Total (cm) |
|-------|--------------|--------------|--------------|---------------|------------|
| 6 | 8.84501 | 6.28639 | 9.78311 | 10.8514 | 14.6103 |

| Count | X error (cm) | Y error (cm) | Z error (cm) | XY error (cm) | Total (cm) |
|-------|--------------|--------------|--------------|---------------|------------|
| 30 | 0.793059 | 0.848473 | 0.868639 | 1.1614 | 1.4503 |

**Figure 4: GCPs locations and error estimates. Z error is represented by the colour of the ellipse. X, Y errors are represented by ellipse shape. GCP locations are marked with a dot. Note that the different scale of the error ellipse in green: in the left image the ellipse in X direction is enlarged 60 times, in the right image it is enlarged 500 times. The reference system is WGS 84 / UTM zone 33N.**

In the last two campaigns we used Theodolite Total Station (TST) to obtain the coordinates of the GCPs. The TST was positioned on the CTN0 point of the local network (Fig.3) which coincides with the roof of the nearby private houses in the northern part (Fig.5A). A classic celerimetric survey was carried out.

The measurements of the GCPs were carried out with a surveying ranging rod equipped with a reflecting prism (offset of -30 mm) assisted by a tripod with a spirit level (Fig.5B), to ensure the upright and stability of the measurement.

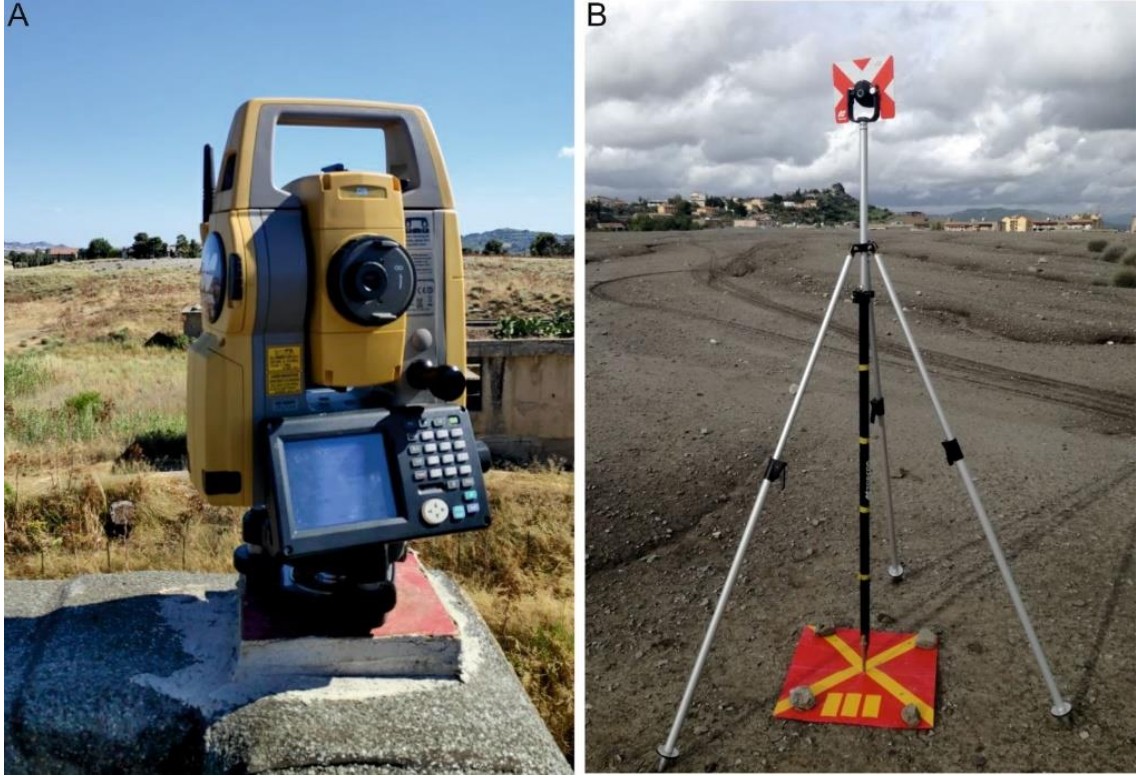

**Figure 5: A) TST DS 103 placed on the fixed metal base in coincidence with point CTN0. B) Surveying ranging rod on the mud volcano during the survey phase.**

We assumed a "Marker Accuracy" on PhotoScan of 5 mm due to the instrumental error. This value has been assumed due to the uncertainty of the CTN0 and CTN1 point coordinates, obtained through GNSS measurements, considering that the uncertainty derived from the TST is negligible.

To validate GCPs data we performed an analysis using the python script "Monte_Carlo_BA.py", with a statistical iterative approach (Monte Carlo approach) (James et al. 2017/a). For clarity, the following terminology will be used in the text:

- GCPs are the points measured in the field. They can be used as control points or check points within the bundle adjustment (James et al. 2017/a).
- Control points are GCPs when they are tied to the model in the bundle adjustment.
- Check points are GCPs when they are not tied to the model in the bundle adjustment.

This script modifies the percentage of GCPs which are used as check points or control points and applies to the check points random variations (James et al. 2017/a). To be more precise values ranging from 10% up to 80% of GCPs have been set.

**2.3 Photo Acquisition**

We have performed five measurement campaigns in approximately one year. The same flight plan was used for all five of them. The AOI (Area Of Interest) was captured by a DJI Phantom 4 Standard, a quadcopter UAV, at a flight height of 33 m above the ground. The sensor size of the UAV's digital camera is 6.17 mm by 4.55 mm, capable of shooting images with a resolution of 12 MP (4000 × 3000 pixels) with a mechanical shutter. Each flight planning was carried out with the Pix4D
Mapper software, adopting a frontal and side overlapping of 80% and 70% respectively. The camera was set-up in a nadir orientation to capture vertical imagery. The flight was carried out in a single grid (simple geometric flight patterns without intersections). An average of 280 images for each survey were acquired with about 1.1 cm Ground Sampling Distance (GSD).

**2.4 Data processing through Structure from Motion (SfM) techniques**

The photogrammetric processing was performed using the commercial software Agisoft PhotoScan. The photogrammetric processing is based on the workflow formulated by the USGS (2017). Steps of the processing chain regarding Tie Point Accuracy and Marker Accuracy have been performed according to James et al. (2017) (Fig.6) .

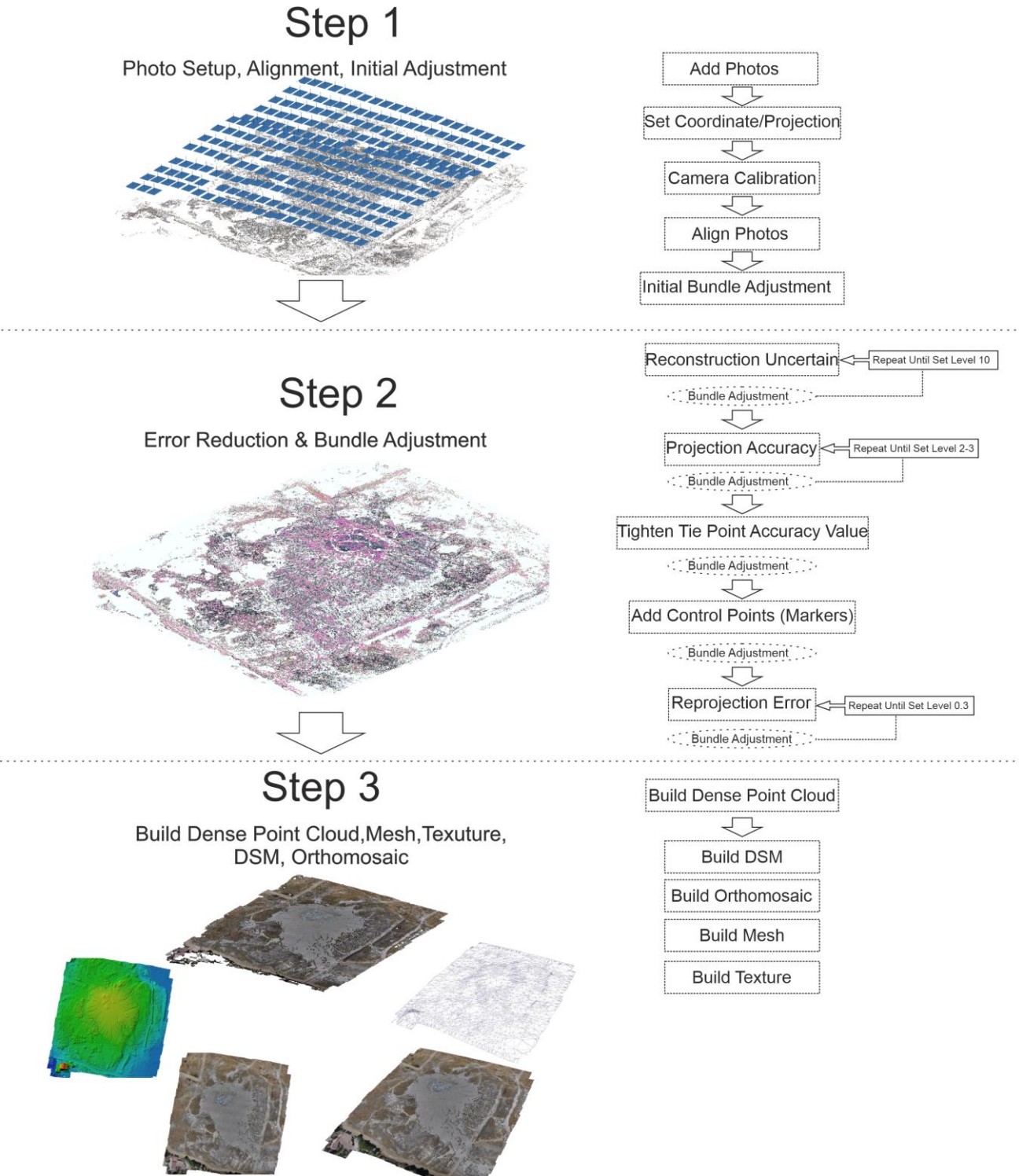

**Figure 6: Simplified block diagram of the photogrammetric processing chain.**


The procedure above is conventional, but we modified some parameters during the cleaning procedure of the sparse point cloud. We adopted the optimal values defined on the workflow (Fig. 6) of "Reconstruction Uncertain" and "Projection Accuracy" in the "Gradual Selection" option in PhotoScan. The "Reconstruction Uncertain" improves the geometric reconstruction of the cloud point. The "Projection Accuracy" improves pixel mismatches in images.

Thus, the obtained cleaned sparce cloud point was tied to the GCPs in order to georeference it.

We perform the "Gradual Selection" option, adjusting the "Reprojection Error" values to reduce the Residual Pixel Errors (Fig. 6) (USGS, 2017)..

Furthermore, appropriate pixel values of tie point accuracy and marker accuracy are set, following the suggestions of James et al. (2017/b).

We applied the "precision_estimate.py" script (James, et al. 2017/b). The script operates, by an interactive Monte Carlo approach, to estimate the accuracy of SfM surveys through photogrammetric and georeferencing parameters, which are then used to provide spatially variable confidence limits for the detection of surface variations.

The precision estimates are calculated through multiple "bundle adjustments" ("Optimize Camera Alignment" in PhotoScan) with different pseudo-random offsets (in this case 4000 pseudo-random offsets) (Fig.7), applied to each image and
checkpoint. The pseudo-random offsets are derived from normal distributions with standard deviations representative of the appropriate accuracy within the survey (James et al. 2017/b).

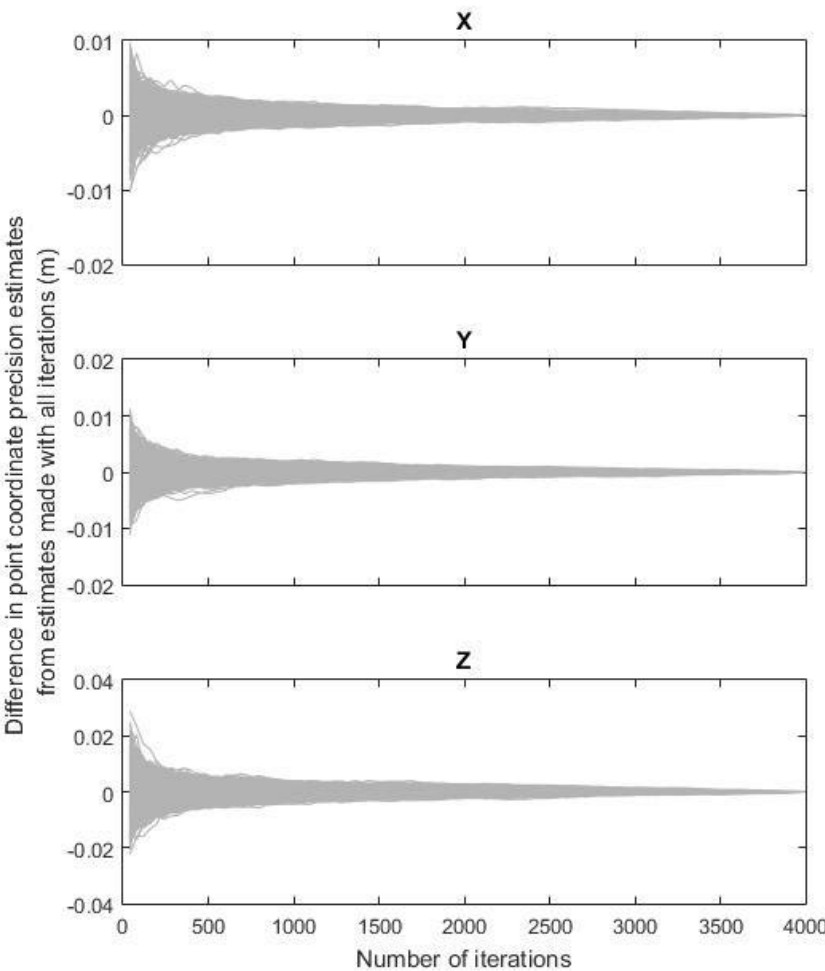

**Figure 7: Example of the iterative approach in the 2020/07/29 survey. Estimates of uncertainties on X, Y and Z improve as number of iterations increase.**

The Sfm-Georef software (James and Robson, 2012) reads the output given by the Monte Carlo python script, setting to each point of the sparse cloud different values of precision on the three spatial components. The results of the script, read by Sfm-Georef, are estimates of the error of the individual points of the sparse cloud point in the three different spatial dimensions (Fig.8).

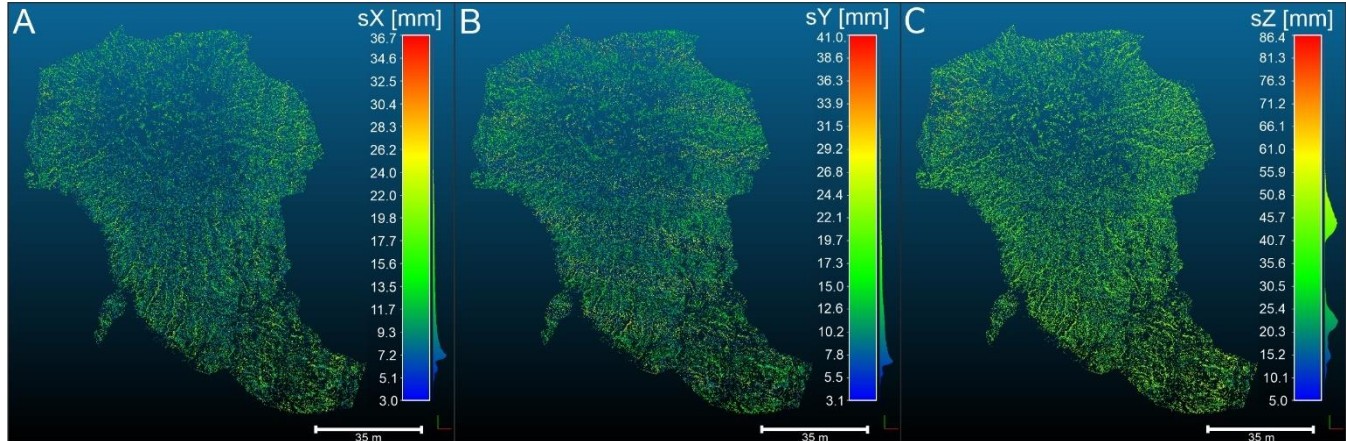

**Figure 8: 3D error estimates of each point of the sparse cloud of the 19/09/13 survey. The error (sX, sY, sZ) is in mm. A) and B) The horizontal errors (X and Y component) are shown. C) The vertical errors (Z component) are shown. The reference system is WGS 84 / UTM zone 33N.**


The 3D topographic change is usually detected from sparse point clouds that have been cleaned to exclude the vegetation that interferes with the comparison techniques. The next step is to link those points to the precision estimates of the sparse cloud; this has been done in CloudCompare v. 2.11.

Through CloudCompare the sparse cloud is interpolated (with the relative precision values on the three-dimensional components) with the dense point cloud. In this phase we decide which interpolation technique is the most suitable, in this case we chose the "nearest neighbors" that consider the three closest points to the sparse cloud, using the median value (better outliner mitigation) to assign the error values to the dense point clouds (Fig.9). This methodology has been chosen due to the heterogeneous distribution of the points in the sparse cloud, avoiding point with null values.

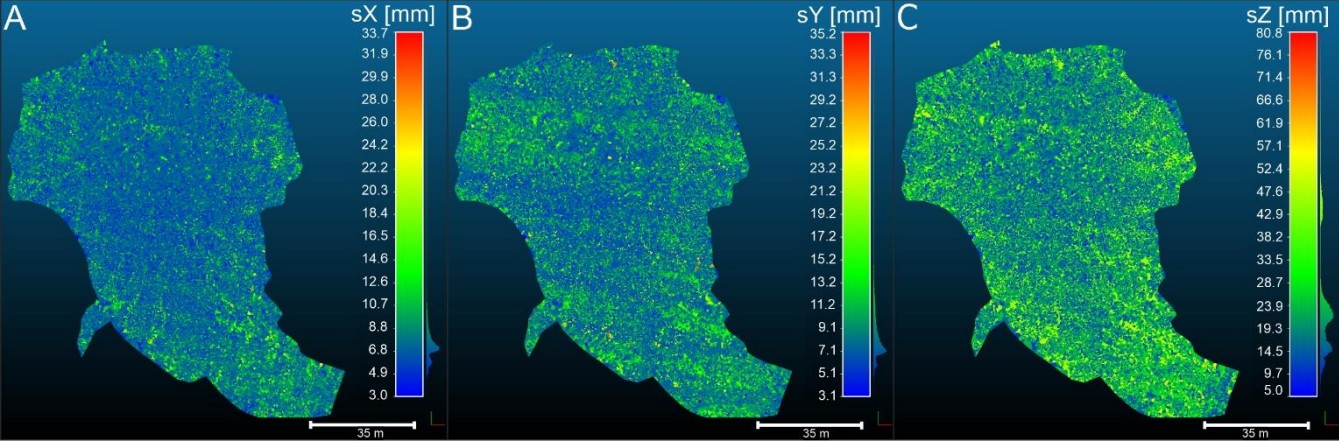

**Figure 9: Estimates of the 3D error of the dense point cloud obtained by interpolation with the sparse cloud of the 19/09/13 survey. The error (sX, sY, Sz) is in mm. A) and B) The horizontal errors (X and Y component) are shown. C) The vertical errors (Z component) are shown. The reference system is WGS 84 / UTM zone 33N.**

Once the precise dense point cloud and their error have been obtained, it is possible to compare the surveys in order to determine the changes between them. Comparisons between the surveys were performed on CloudCompare by the M3C2-PM plugin (James et al., 2017/b; Lague et al., 2013) that identifies a statistically significant change where the topographical differences exceed a value of spatially variable uncertainty. According to James et al. (2017/b), the M3C2-PM is particularly suited for point clouds derived from SfM. The M3C2-PM (James et al., 2017/b) uses estimates of precision of the

coordinates of points (3D precision map) that we have previously calculated.

The outputs of M3C2-PM are scalar values applied to the cloud:

- significant change (Fig.10A).
- M3C2 distance (Fig.10B).
- distance uncertainty (Fig.10C).

The first output (Fig.10A) shows the changes which exceed the uncertainty values in both point clouds. It represents a confidence interval constrained by values of Level of Detection $_{95\%}$ (LoD$_{95\%}$) which are spatially variable. This is applicable in any morphological setting, providing a reliable 3D analysis of topographical change.

The second output (Fig.10B) shows the calculated distances between the two clouds.

The third output (Fig.10C) shows the uncertainty values of the distances and their spatial variation between the two clouds.

Once the uncertainty value is defined, the changes are significant when they overtake the value of the uncertainty.

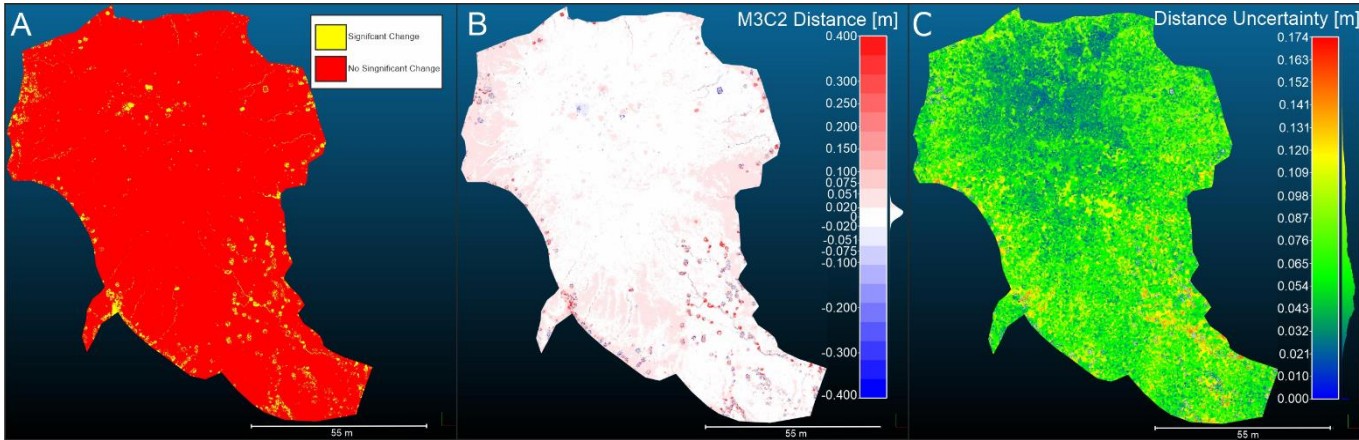

**Figure 10: Point clouds resulting from M3C2-PM processing, comparison between 2019/07/29 and 2020/06/15; (A) significant changes between the two point clouds with LOD$_{\%95}$; (B) distances between the two point clouds; (C) uncertainty of the distance between the two point clouds. The reference system is WGS 84 / UTM zone 33N.**

In the 2020/06/15 survey, we used callipers to test and validate the method of the measurements. Five numbered callipers, with steps of increasing height of about 2 cm, were positioned on the mud dome. The heights are 2, 4, 6, 8 and 10 cm.

These were used to obtain an instrumental sensitivity of the measures (Fig.11). All callipers are detected, we show as example the smallest (Fig.11).

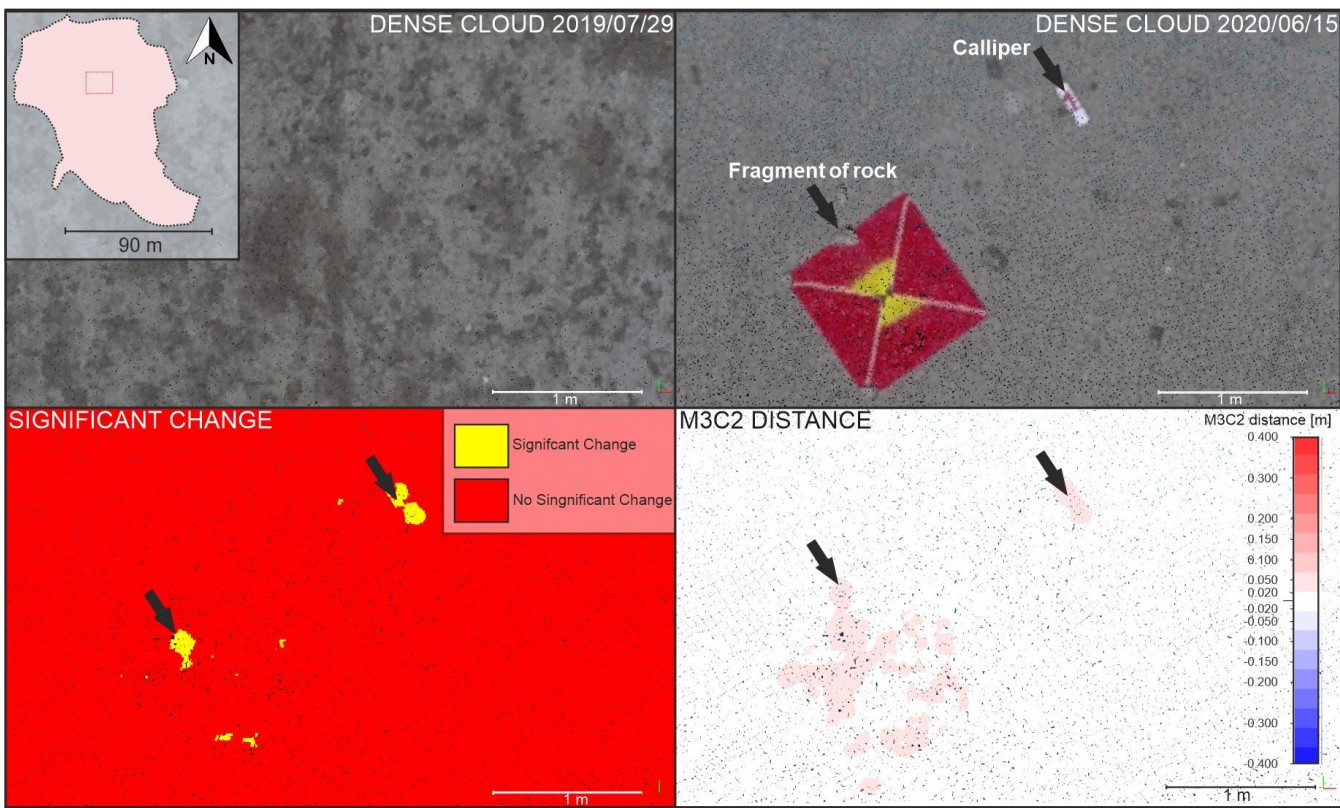

**Figure 11: On the upper part of the figure the dense point cloud of the two surveys of the same area carried out on 2019/07/29 (left) and 2020/06/15 (right) are shown. On the lower part the clouds with the significant change (left) and the M3C2 distance (right). The arrows indicate the significant changes between the surveys. The calculated changes between the two clouds estimate an altitude increase of about 6.3 cm for the fragment of rock used to maintain the target in position and an increase in height of about 2.5 cm for the calliper which is 2 cm thick. The reference system is WGS 84 / UTM zone 33N.**

## 3. Results

As illustrated in section 2.2, the results show that using between 40 and 60 % of the GCPs (as control points) the RMSE value has minimal variation. Thus, the optimal minimum number of GPCs is between 12 and 18. When the threshold of 60% is exceeded, there are no significant improvements (Fig.12). This result has been confirmed in all campaigns.

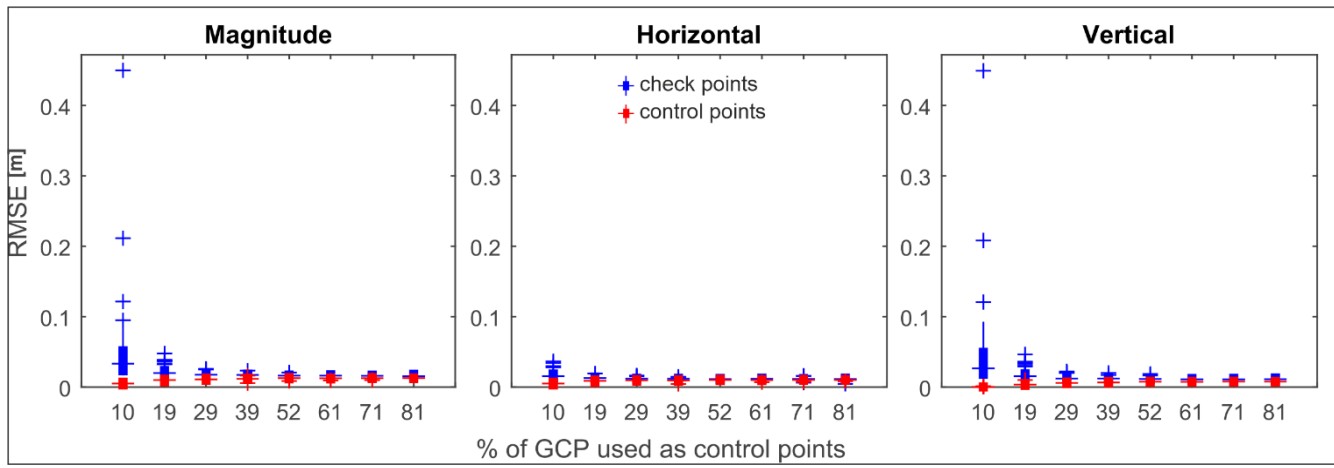

**Figure 12: The boxes represent the distribution of the RMSE of GCPs (2019/09/13 survey) on the three components: Magnitude (3D), Horizontal and Vertical according to the % of the GCP used as control points. RMSE is calculated on 50 self-calibrating bundle adjustments for each % of GCPs used as control points. The GCPs are randomly selected for each self-calibrating bundle adjustments. The central bars indicate the median RMSE values, which are included in the boxes that extend from 25th to 75th percentile and the outliers are indicated by the + symbols.**

The results obtained from the photogrammetric comparisons supported by geodetic topographic survey, have an average uncertainty of about 6.4 cm, with a minimum of about 2 cm and a maximum of about 12 cm, relative to an area of 42700 m² (Fig.13A). The uncertainty of the central area has an average value of about 3.9 cm, with a minimum of about 2 cm and maximum of 10 cm, on an extension of 360 m² (Fig.13B).

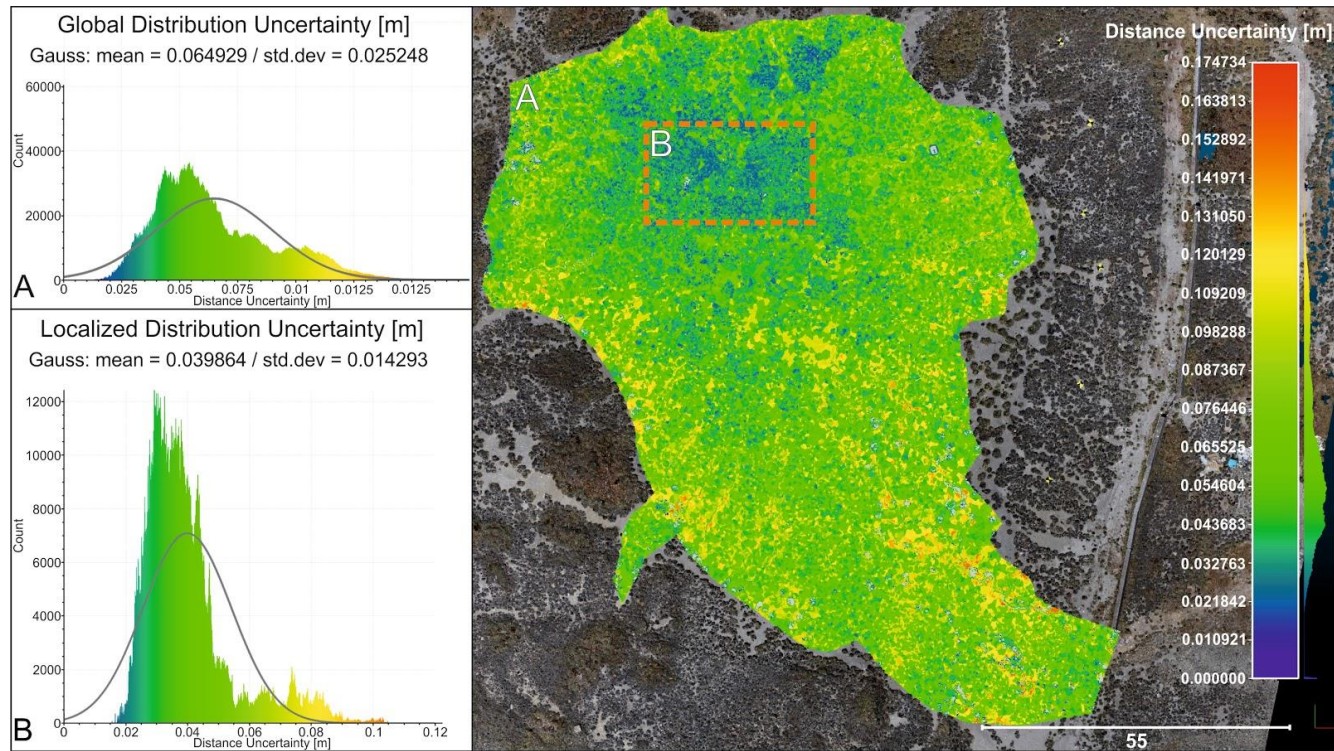

290 **Figure 13: Points Cloud with uncertainty distance values between 2019/07/29 and 2020/06/15 (right). On the left, (A) the Gaussian distribution of the distance uncertainty over the whole area is shown, the mean is about 6.4 cm and the standard deviation is about 2.5 cm. (B) the Gaussian distribution of the distance uncertainty of the central emission area is shown, the mean is about 3.9 cm and the standard deviation is about 1.4 cm. The reference system is WGS 84 / UTM zone 33N.**

295 During the monitoring period, in addition to natural changes, we recorded anthropogenic action, due to the relocation of objects (garbage) in the area after unauthorized access. These changes have a decimetric order of magnitude and are easily detected by the technique used (Fig.14).

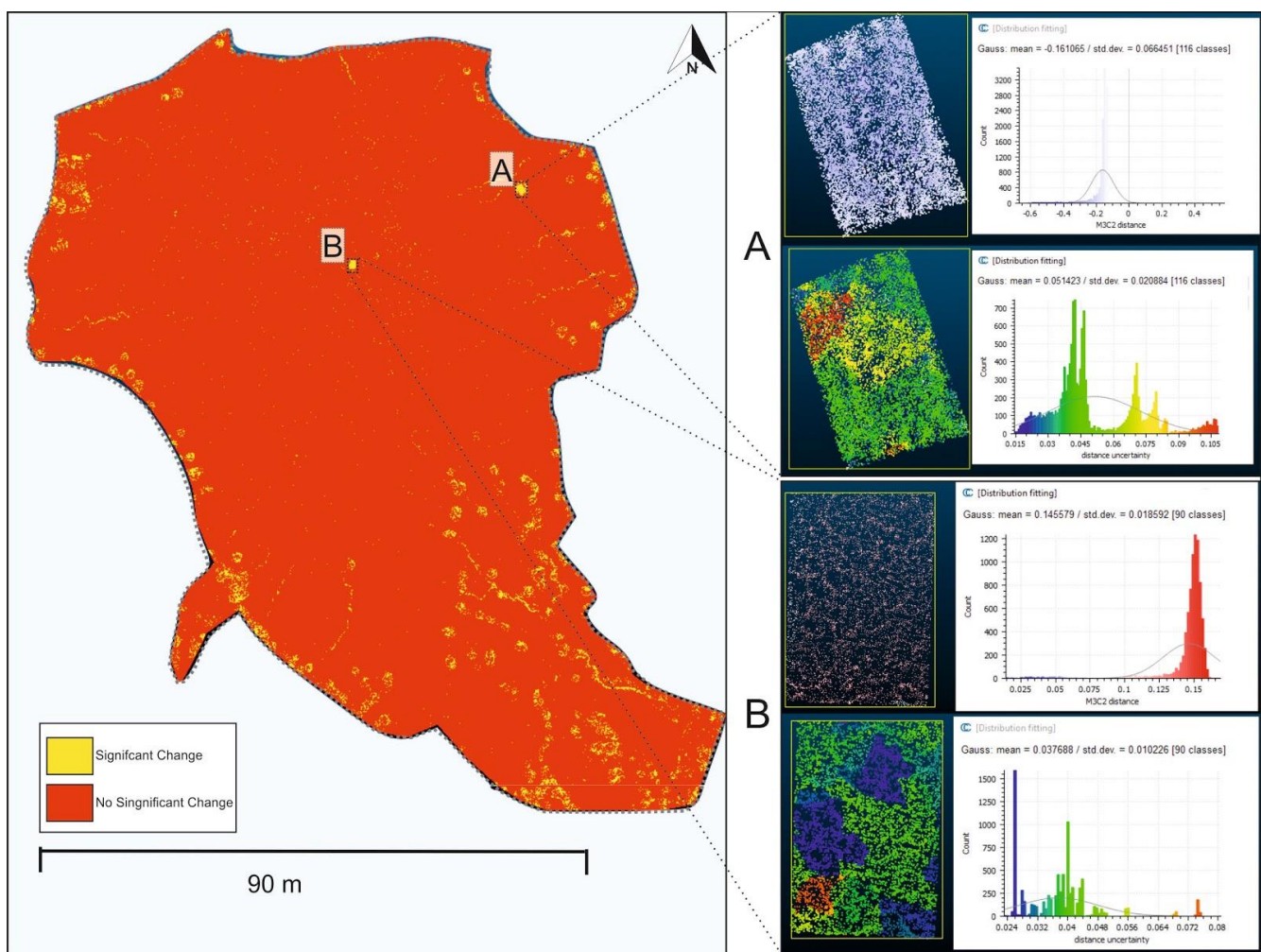

Figure 14: Enlargement of the comparison between the point clouds of the 2019/09/13 and 2019/10/14 campaigns with significant change. Two significant changes (A and B) are shown, which highlight an anthropogenic action, i.e. the movement of a wooden platform about 15 cm high. (A, upper part) a lowering of an average height of 16 cm is detected, the Gaussian distribution of the M3C2-PM distances with the mean and the standard deviation are shown. (A, lower part) the Distance Uncertain varies spatially with an average value of about 5 cm, the Gaussian distribution of the M3C2-PM distances with the mean and the standard deviation are shown. (B, upper part) an increase of height of 14 cm is recorded, the Gaussian distribution of the M3C2-PM distances with the mean and the standard deviation are shown. (B, lower part) the distance uncertain has an average value of about 3 cm, the Gaussian distribution of the M3C2-PM distances with the mean and the standard deviation are shown. The reference system is WGS 84 / UTM zone 33N.

Two types of analysis were carried out: semi-quantitative and quantitative. The surveys 2019/07/29 and 2020/06/15 (time interval of about 1 year) were chosen to perform the semi-quantitative analysis. This analysis was carried out on the whole mud volcano area, the objective is to detect significant deformations (in the order of decimetres). In order to visualize this deformation (Fig.15), the values of M3C2 distance ranged between -2 and 2 cm were excluded, according to the minimum value of distance uncertainty. This range was verified instrumentally with the use of callipers.

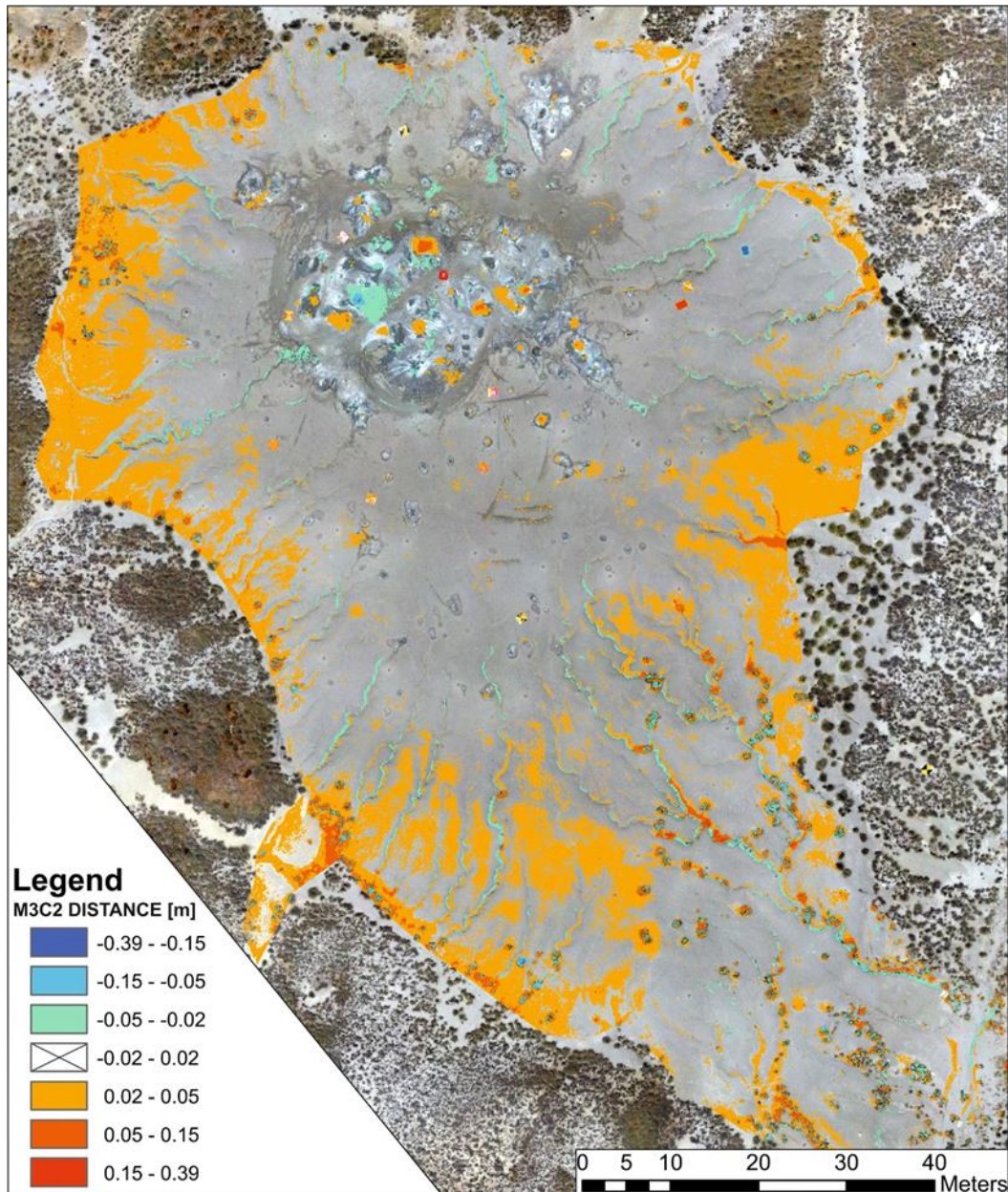

**Figure 15: Orthophoto with M3C2 Distance. The distance scale has been adjusted to exclude values between -2 and +2 cm. The positive distances, observed at the margin of the mud dome, are interpreted as mud flow or sediment deposition coming from peripheral griffon vents eruptions or erosion of summit area. The reference system is WGS 84 / UTM zone 33N.**

320

Comparing the 2019/07/29 and 2020/06/15 surveys (Fig. 15), we observe that the surface of the volcanic cone is not affected by deformations. Only morphological changes in the volcanic structures can be underlined, such as small eruptive cones, griphons, sauces and mud pools.

The quantitative analysis was performed on small central portions of the mud volcano. The aim of the quantitative analysis is to estimate the trend and evolution of the deformation. To assess the deformation and the local morpho-structural evolution, two temporal series have been developed in two areas: Z1 (collapse zone) and Z2 (uplifting zone) (Fig.16 and Fig.17). The zones were chosen related to the significant change. On the selected zone, the average distances between points with significant change were computed by M3C2-PM. Z1 has an extension of 1 m² (Fig.16) and Z2 has an extension of 0.84 m² (Fig.17).

As master was chosen the campaign of 19/07/29 (T0) (Table 2).

| $T_0$ | 2019/07/29 |
|---|---|
| $T_1$ | 2019/09/13 |
| $T_2$ | 2019/10/14 |
| $T_3$ | 2020/01/13 |
| $T_4$ | 2020/06/15 |

**Table 2: Surveys used to the generation of time series.**

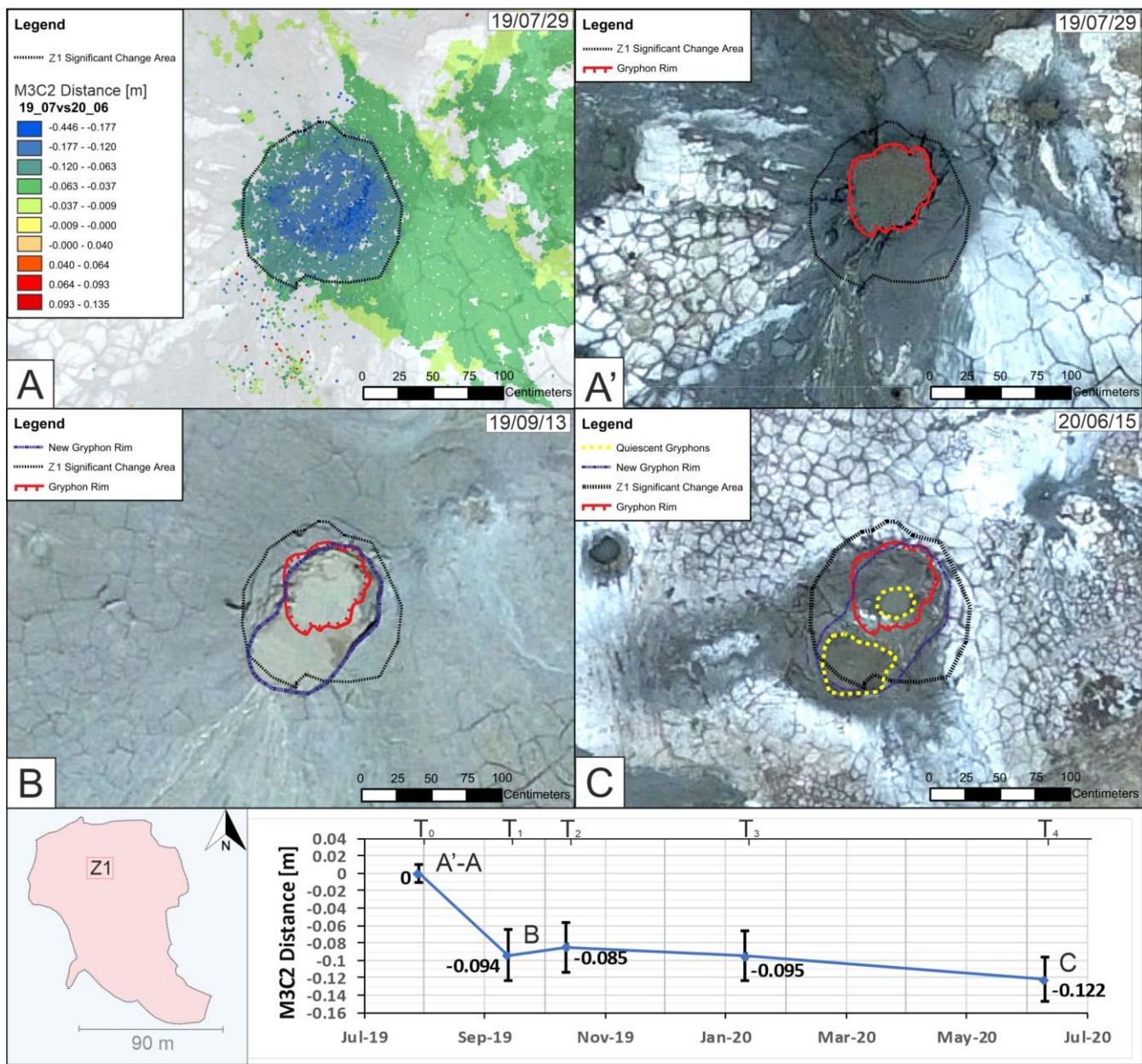

**Figure 16: (A)** M3C2 Distance (2019/07/29 vs 2020/06/15) of Z1; the area used to create the time series is delimited with a dashed line (selected by significant change). **(A')** Orthophoto of 2019/07/29 with the gryphon border in red. **(B)** The orthophoto of 19/09/13 highlights the expansion of the gryphon border (blue). **(C)** Orthophotos of 2020/06/15, gryphon during the quiescence phase and the split of the channel in yellow. Below the time series with a decreasing trend is shown. The reference system is WGS 84 / UTM zone 33N.

In Figure 16 the subsidence (about 10 cm / 60 days) of gryphon is represented by the southwest migration of the mud pool edge. In Figure 17, the uplift (about 4 cm / 60 days) of a blind gryphon is represented by the development of radial fractures on the cone surface. Close to the growing gryphon, another indication of deformation is the deviation of a mudslide flowing in the N-S direction towards the southern sector of the analysed area. Finally, the sudden appearance of the gryphon is well highlighted in the ortophoto of the last survey (Fig.17).

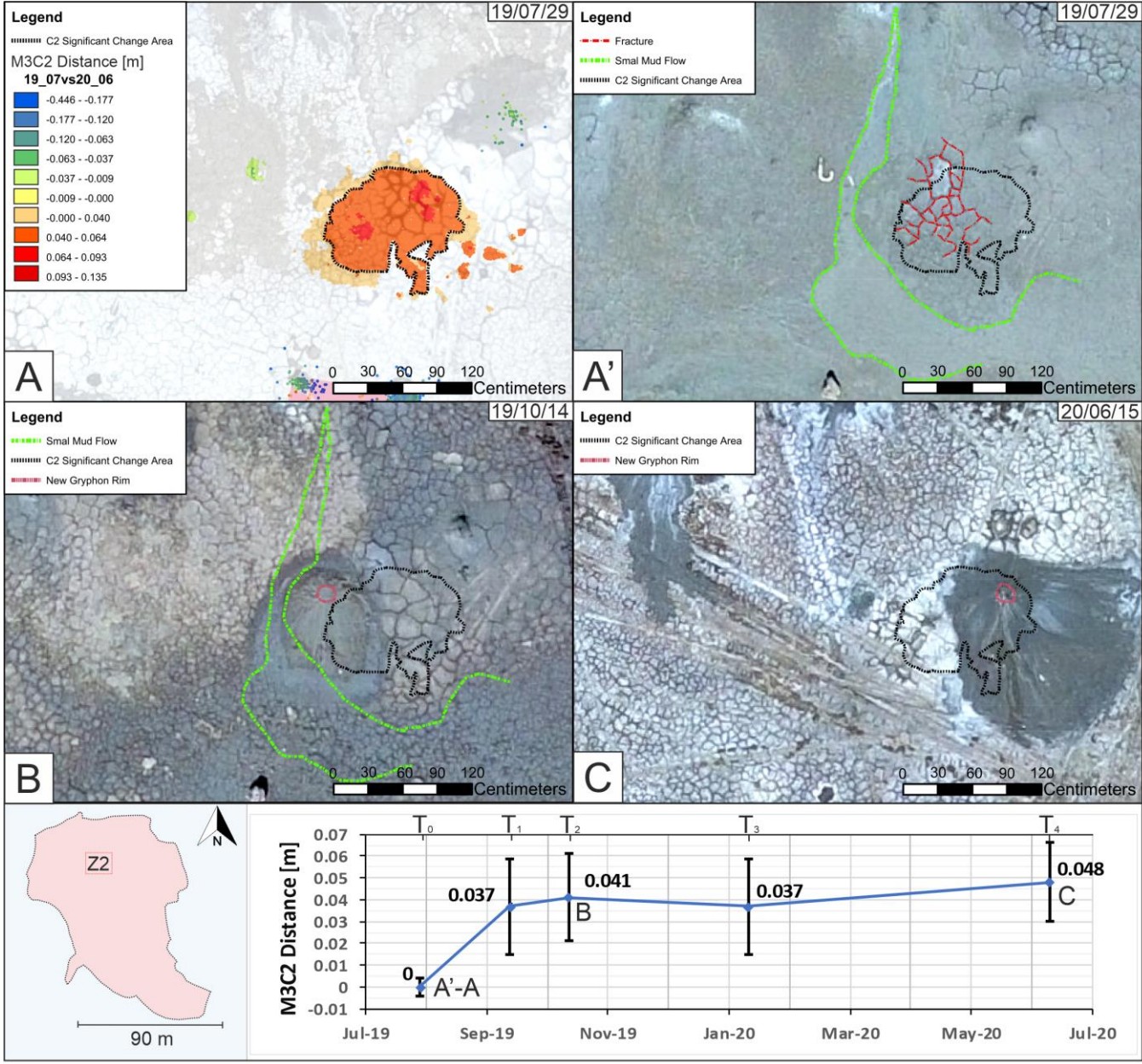

**Figure 17: (A) M3C2 Distance (2019/07/29 VS 2020/06/15) of the Z2, the area used to create the time series is delimited with a dashed line (selected by significant change). (A') Orthophoto of 2019/07/29, the margin of the deflected flow, in green, and the radial fractures, in red. (B) Orthophoto of 19/10/14, formation of a new emission gryphon in red and the previous flow in green. (C) Orthophoto of 2020/06/15, formation of a new emission gryphon, in red. Below the time series with upward trend is shown. The reference system is WGS 84 / UTM zone 33N.**

## 4. Discussion

UAV technology combined with SfM is a valuable tool in geological risk assessment and monitoring, however, some issues must be considered.

The first is the quantity and quality of the acquired GCPs. According to the results, the minimum optimal number for geo-referencing is 12 GCPs. A slight improvement is observable up to 18 GCPs, while no appraisable improvement is detected for a higher number of GCPs (Fig.12). In addition, the method of acquiring GCPs reduces their error by using the Total Station Theodolite (Tab.1). The combined use of high-precision topographic instruments with an optimal number of GCPs improves the reliability of the datasets.

The second aspect to be considered is the evaluation of distance uncertainty when two surveys are compared. The distance uncertainty between two data sets (surveys) can be considered as an estimate of the sensitivity of the methods to detect measurable topographic changes. The results of the M3C2-PM show that the average distance uncertainty between the first and the last surveys is about 6.5 cm over the whole area at the 95% confidence level (Fig.13A). Furthermore, considering a smaller portion of the area, the uncertainty decreases to about 3.9 cm at the 95% confidence level (Fig.13B). This allows us to analyse certain morphological changes and anthropogenic activity on the mud surface (Fig.14). The anthropogenic activity determined a height decrease of about 16 cm where a wood platform was previously located. As well as, a height increase of about 14 cm is recorded at the place where the wood platform has been relocated. The values recorded are consistent with the real thickness of the object detected, differing by 2 cm (Fig.14). The callipers show that it is possible to measure changes at least of 2 cm (Fig.11).

After the uncertainty and sensitivity of the surveys were computed, the time series were made. Considering the significant changes and their errors, we computed the trend of deformation of two areas in order to reconstruct the evolution of the phenomena which generated these changes (Fig.16 and Fig.17).

Considering the results obtained by SfM, we propose a monitoring system of Santa Barbara mud volcano based on a semi-quantitative approach. We defined three states by setting a space-time range:

- Normal state if the deformation value does not exceed 5 cm in the range between 1 to 2 months.
- Pre-Alert state if the deformation value is between 5 cm and 10 cm during a period of between 1 to 2 months.
- Alert state if the deformation value exceeds 10 cm in the range between 1 to 2 months.

. To define the state of the activity of the mud volcano, the area affected by deformation must be a significant area of the total surface of the volcano. The definition of the activity is an important aspect of the monitoring of hazard, because there are many natural phenomena that can produce deformation in a small area (like a gryphon) or in a larger portion (like the sedimentation occurred at the border of the mud volcano). The significant area affected by deformation must be correctly defined by an operator (public agency, university, institute of research or Dipartimento della Protezione Civile) which really

knows the natural phenomena that can occur on the Santa Barbara mud volcano.

The methodological approach is valuable and efficient from the point of view of quantity and quality of data collected in relation to the work and time spent. This monitoring  techniques is a useful tool to detect the early unrest phase of the mud volcano usually induced by changes in pressure and volume of fluid raising from the stagnation chamber.

The pre-eruptive deformation consists of a marked uplift and occasional small subsidence which are probably related to the

redistribution of the subsoil of the pressurized fluids (Antonelli et al. 2014). According to Antonelli et al. (2014), soil uplift can occur up to a year before the eruption.

## 5. Conclusion

In hazard management, the SfM technique (Gomez et al., 2016, Kaab, 2000, Fugazza et al., 2018; Giordan et al., 2017, 2018) starts to be largely used by the scientific community. In the monitoring of potentially dangerous active sites, the UAVs are

very advantageous because they are not used only as a support in the post-disaster events (Rokhmana and Andaru, 2017; Hisbaron et al., 2018), but also for the pre-event monitoring.

According to Kopf (2002), Antonelli et al., (2014), Madonia et al., (2011), INGV (2008a) and Regione Siciliana (2008), the deformations of the surface shell of mud volcanoes can occur up to one year before the paroxysmal event with doming and development of structural lineaments with order of magnitude from centimetres to decimetres.

The results allow us to define the criteria for monitoring and analysing the study area. For the mud volcano of Santa Barbara, the monitoring criteria are:

- Monitoring interval between 1 and 2 months.
- The optimal number of GCPs is between 12 and 18.
- Acquisition of GCPs by high-precision topographical instrumentation (TST).
- The processing chain of the sparse point cloud according to workflow of USGS (2017) was enhanced by the correct value of "Tie Point Accuracy" and "Marker Accuracy" as suggested by James et al. (2017).
- An assessment of the state of activity of the mud volcano based on a semi-quantitative approach.

The frequencies of the campaigns depend on the status of activity, while the other criteria depend on the object / structure of the monitoring.

These criteria allow us to detect events with deformation of at least 2 cm. In the case of anomalous values detected, the monitoring campaigns must be improved. This involves extensive monitoring, such as: i) developing time series localized in key areas and ii) combining different methodologies, e.g. micro-seismicity monitoring and three-dimensional geophysical prospecting (Imposa et al., 2016) to improve the monitoring system of the active geological process.

## Acknowledgements

This paper was carried out with the financial support of the Presidenza del Consiglio dei Ministri (Project Code E98D19000000001, Monitoraggio e Studio dei Processi di Deformazione Superficiale Connessi al Vulcanismo-Sedimentario delle Maccalube di Santa Barbara (Caltanisetta) within the DPCM 25th May 2016 - Riqualificazione Urbana e la Sicurezza delle Periferie delle Città Metropolitane, dei Comuni Capoluogo e delle Città di Aosta, Scientific Supervisor: G. De Guidi). We thank Dr. Silvia Rita Popolo for her valuable support.

Finally, we are very grateful to the two anonymous referees who helped us to enhance the clarity and structure of the content of this manuscript.

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
