# Peer review of "UAV survey method to monitor and analyze geological hazards: The Case study of the mud volcano of Villaggio Santa Barbara, Caltanissetta (Sicily)"

_Natural Hazards and Earth System Sciences, 2020_

## Referee Comment (RC1) · Anonymous Referee #1 · 7 Jan 2021

Suggestions for revision or reasons for rejection

A pdf report with color coding for better understanding of the comments is attached.

1. General Comments Hereby I want to state that I reviewed this manuscript to my best ability and approached it with interest and with the intention to give suggestions which improve the quality of the work. The methods used in the paper are interesting and relevant for future studies in order to optimize SfM methods for ground deformation analysis and monitoring. However, the paper needs reorganization, clarification and

rephrasing. I want to address that the presented manuscript does not contain a Results chapter but instead goes directly from Methods to Discussion. However, the results of the study are presented in both Methods and Discussion. This makes the article confusing and resulting data unclear. I would strongly recommend a reorganization of the content with a clearer separation between Methods, Results and Discussion. Doing this will increase the readability of the work significantly. Furthermore, the first paragraph of the Conclusion would make more sense in the Introduction chapter. I would as well strongly recommend having this paper proof-read by a native English speaker before the next submission to correct and improve the phrasing. The sentence structure is sometimes too complicated or incorrect so that I think the paper quality would largely benefit from having it proof-read.

2. Specific Comments Abstract Line 12- 15: "Among all the active geological processes on Santa Barbara mud volcano (Caltanissetta town, Italy), represents a dangerous site because it caused, on 11 August 2008, a paroxysmal event, which determined severe damages to the infrastructures at around to 2 km the paroxysmal event." confusing - rephrase this sentence Introduction Line 40 – 44: "The accuracy of 3D information can be significantly increased by the ground control points (GCPs) by georeferencing the data to ground control points (GCPs). The GCPs acquisition acquisition is a undamental fundamental aspect of the georeferencing of the network of images captured by UAV photogrammetry – unclear phrasing. Images are not captured by UAV photogrammetry. Images are captured "for photogrammetry" or "by UAV". In this process a right number of GCPs is required which lead to a greater accuracy of the outcomes (point clouds, 3D grid, orthomosaic or digital surface model (DSM))." - rewrite these sentences more concise without repetition. e.g.: "Ground control points (GCPs) are used to improve the accuracy of the resulting data. Therefore, points recognizable on the UAV imagery are measured with a survey device to georeference the data." Line 46: "In this paperthrough SFM,. . ."- what do you mean? As well use a consistent acronym for SfM Line 70 – 73: "This morphometric structure is typical of uplifting areas and therefore relative decrease of the base level. This morphometric evidence suggests uplift processes- repetition of the canopy volcano area- what is that?? that lies above a stagnation chamber that has been carried out for this research project through geophysical investigation- confusing, rewrite this sentence." Methods Line 98: Figure 3 Incomplete Legend -what are the other colors and symbols in the map? -probably choose different background map. -what is the CRS /grid? Line 117-119: "During the monitoring sessions there has been ongoing research on the best Ground Control Points (GCPs) methods of acquisition. This was aimed at a more detailed indication of the 3D deformation process of the volcanic cap and a vertical and horizontal geometrical resolution of centimetre/subcentimetric order of magnitude." – confusing. Rewrite those sentences Line 116 and onwards: Ground Control Points (GCPs) Please explain the different errors you mention and how they are calculated. Explain "error", "total error", "GCP error" and "average total 3D error" As well clarify for the reader what the difference between "GCPs", "Control Points" and "Checkpoints" are. Line 122: "10-1 :10-2 m"- please clarify what this means Line 127-128: confusing, statistical analysis of what? The whole sentence should be clarified Line 129-130: "Since 2019 only TST has been used for the GCP survey and according to Tahar et al. (2013) the number of GCPs has been increased (Tab.1) reducing the error to $\approx$ 1.4 cm and in the last campaign $\approx$ 0.7 cm." -are you saying the increase of the GCPs has reduced the GCP error or the fact that only TST was used for the survey? please clarify. Line 131: "...greater control.."- what do you mean by that? – higher accuracy? Line 131:"... limit number..."- explain what you mean by this. Figure 4: it would be beneficial to have a relative scale of the horizontal error below the green ellipse e.g.: (in X- direction) or something similar. Line 151-152: "Photoscan accuracy" – what do you mean? The accuracy of the resulting data? Clarify Figure 6: what do red and blue mean? Line 164-165: "25° to 75° percentile" – do you mean 25th to 75th percentile? Photo Acquisition Line 173-174: "The camera was oriented in a 90- degree angle"- can be confusing for the reader. Say that "vertical imagery" was acquired Line 174: explain what a single grid is. Line 174: Introduce the acronym (ground sampling distance) Data Processing SfM Line 189-190: "value" doesn't seem the right term here. Rather use step/s of

the processing chain Line 203: "measurements of check" -checkpoint measurement?? Line 215: which version of Cloud Compare? Line 219-220: "This methodology has been chosen having in mind the heterogeneous distribution of the points in the sparse cloud, avoiding holes and thus null values." - is this why you chose this interpolation method or is it just something you keep in mind? - clarify and replace the term "having in mind" with something more precise. eg.: "This method has been chosen "because of"/ "due to" the heterogeneous distribution...." Line 221: which "measurements" do you mean? Line 223-224: it would be beneficial to include an illustration of a precision map Figure 9: Numbers on legends too small Figure 11 and Line 264-265: "These were used to have an instrumental sensitivity scale of the measures of figure 12. The first and lowest one (ËIJ2 cm) was easily detected." - what about the others? up to 10 cm? or are you saying that if only one is easily detected the others are not necessary. If you are not going to talk about the other calipers it should be considered whether figure 11 is necessary. Discussion Line 289: "....temporally distant peaks...." – clarify what you mean by this? Data sets? Line 299: clarify in the caption which M3C2 distance is shown here (between which datasets) - explain what processes lead to the M3C2 distance around the edges of the survey area? or is this due to lower precision and/or accuracy in these areas? Figure 15 and 16: add error bars to the time series. Increase the size of the font on the scale Conclusion Line 330-341: this paragraph and references should be part of the introduciton. Line 351: clarify what kind of interval you mean by "between 30 and 60 g" Line 352-353: What does this mean for other projects? Is this amount of GCPs scalable for different area sizes?

3. Technical Comments Here are I point out minor technical suggestions with the same color coding as before. Line 1: write "monitoring" with lower case m Line 2: "the case study of "a"/"the" mud volcano Line 16: replace "danger" with "hazards" Line 18: "for monitoring of deformation processes..." Line 21: Introduce abbreviation SfM Line 34: "The different acquisition methods...." Line 35 - 36: "In this context, UAVs therefore offer unprecedented spatial and temporal resolution...." - repetition of the previous sentence. Line 38: "…..Lidar, thermal imaging cameras,….." – add the comma Line

44: "... is also affected by other features, for example...." –insert: controlled by other variables, such as:.... Line 44 -45: "design and altitude of the flight" flight path and flight altitude Line 48- 49: 95% (LoD 95%) Level of Detection - 95% Level of Detection (LoD 95%) Line 55: "Ortho-photos generated by UAVs in the area of the Santa Barbara mud volcano."- Ortho-photo of the Santa Barbara mud volcano generated from UAV imagery Line 58: "....developed since Late Miocene untill the Quaternary" – "...which developed from the Late Miocene to the Quaternary,.... Line 59: ".....formed by a foreland fold and a thrust belt...." Line 60: "....the clastic sediments deposited...."- "...the deposition of clastic sediments..." Line 60-61: "....during the late Miocene towards the Pleistocene...."- from the late Miocene to the Pleistocene Line 65: "On the mud volcanoes..."- which mud volcanoes?? Line 62: "More hover...." ? Line 73. Personal communication with who? Which methods where used? Line 73: "sill-like" Line 74: replace develops with "sits" or "is located" Line 74: 50 meters Line 81: "On the surface outcrops deformation structures ..." – rewrite Line 82-83:"... highlighting what high intensity of stress and strain the volcano can generate..." ... highlighting the high stress and strain environment ..... Line 84-85:"....and we still believe they are active...." .... and are still believed to be active, Line 91: monitoring – monitor Line113: introduce acronym for TST Line 113: "...TST base local station..." – local base station / local TST base station??? Line 120: Real Time Kinematics (RTK) Line 121: GCPs Line 121: "...(Tab. 2).." – do you mean Table 1?? Line 125: "detected"-"measured" Table 1: "GCPs NUMBER" – GCP NUMBER Line 135: "...on the table..." – in the table Line 141: rephrase the sentence. Line 152: "...CTN1 points coordinates..." Line 175: "... a cheap UAV.." – redundant information. Rather mention that it is a quadcopter UAV Line 175: ".... Fly flight altitude......" Line 176. "The feature of...."- feature is the wrong term here. Just say: The sensor size is..... Line 193: add """ to the Gradual Selection Line 196-197: James, et al....... Line 210: "Secondly,..." where is the firstly?? Line 248: significant change Line 275: "...survey techniques, they have a very...." Line 289: "We....." Line 290: "Data are still....."- "The data is still .... Line 294: "In literature..." – "According to literature...." Line

360: "In this case more thoroughly monitoring will carried out:. . ."- rewrite this sentence.

Please also note the supplement to this comment:
https://nhess.copernicus.org/preprints/nhess-2020-378/nhess-2020-378-RC1-supplement.pdf

───────────────────────────

---

## Referee Comment (RC2) · Anonymous Referee #2 · 20 Jan 2021

- The list of references in the introduction section must be updated with recentmost researches (2018-2020), they are many.

- The text regarding the GCP in the section must be moved to the methods.

- Figure 1 show the orthomosaic resulting from photogrammetry processing. It can be moved to the results section. A geological introduction to the area is mandatory.

- Report the reference system in all figures.

- There is a lack of references in the methods as well as there is a lack of details regarding the processing. I suggest reading recentmost papers focusing on the methodology.

- It has been generated a DSM (Digital surface model) not a DEM.

- Conclusion must be rewritten removing the lines listing several references that can be added to the introduction section. Conclusions must address the main results from the present work.

---

## Referee Comment (RC3) · Anonymous Referee #2 · 22 Jan 2021

- I suggest an improvement for the literature, some recent and important paper are missing.

- The methodology must be improved, both regarding the UAV survey and SfM processing.

- Figure 1 cannot be the Orthomosaic resulting from processing, it is a result.

- Conclusions must be rewritten avoiding the long list of citation.

- Add the reference system in all figures.

---

## Short Comment (SC1) · Fabio Brighenti et al. · 10 Feb 2021

Reply to Reviewer 1

UAV survey method to Monitoring and analysing geological hazards: The Case study of mud volcano of Villaggio Santa Barbara, Caltanissetta (Sicily)

Fabio Brighenti et al.

A pdf file of the reviewed manuscript is attached as supplement file. The changes in

the text are marked with green for rephrasing, reorganised and new parts added, and with yellow the changes suggested by the referees.

1. General Comments Hereby I want to state that I reviewed this manuscript to my best ability and approached it with interest and with the intention to give suggestions which improve the quality of the work. The methods used in the paper are interesting and relevant for future studies in order to optimize SfM methods for ground deformation analysis and monitoring. However, the paper needs reorganization, clarification and rephrasing. I want to address that the presented manuscript does not contain a Results chapter but instead goes directly from Methods to Discussion. However, the results of the study are presented in both Methods and Discussion. This makes the article confusing and resulting data unclear. I would strongly recommend a reorganization of the content with a clearer separation between Methods, Results and Discussion. Doing this will increase the readability of the work significantly. Furthermore, the first paragraph of the Conclusion would make more sense in the Introduction chapter. I would as well strongly recommend having this paper proof-read by a native English speaker before the next submission to correct and improve the phrasing. The sentence structure is sometimes too complicated or incorrect so that I think the paper quality would largely benefit from having it proof-read.

Reply: Thank you for your interest in understanding and helping us with the paper. We want to thank you for your kind advices. The paper has been reorganised; the chapter Results has been added. The paper now follows the order of the chapters Methods, Results and Discussion. The data has been reordered within the Results chapter. Large parts of the paper have been reorganised, clarified and rephrased. The first paragraph of the Discussion has been moved to the Introducion. The paper has been corrected by a native speaker.

2. Specific Comments

Abstract Line 12- 15: "Among all the active geological processes on Santa Barbara mud volcano (Caltanissetta town, Italy), represents a dangerous site because it caused, on 11 August 2008, a paroxysmal event, which determined severe damages to the infrastructures at around to 2 km the paroxysmal event." Confusing - rephrase this sentence.

Reply: The sentence has been rewritten (Line 15-17).

Introduction Line 40 – 44: "The accuracy of 3D information can be significantly increased by the ground control points (GCPs) by georeferencing the data to ground control points (GCPs). The GCPs acquisition acquisition is a undamental fundamental aspect of the georeferencing of the network of images captured by UAV photogrammetry – unclear phrasing. Images are not captured by UAV photogrammetry. Images are captured "for photogrammetry" or "by UAV". "In this process a right number of GCPs is required which lead to a greater accuracy of the outcomes (point clouds, 3D grid, orthomosaic or digital surface model (DSM))." - rewrite these sentences more concise without repetition. e.g.: "Ground control points (GCPs) are used to improve the accuracy of the resulting data. Therefore, points recognizable on the UAV imagery are measured with a survey device to georeference the data."

Reply: The sentences were rewritten and clarified (Lines 53-54).

Line 46: "In this paperthrough SFM,. . ."- what do you mean? As well use a consistent acronym for SfM Reply: There was a typo in the sentence. The acronym is inserted (Line 20).

Line 70 – 73: "This morphometric structure is typical of uplifting areas and therefore relative decrease of the base level. This morphometric evidence suggests uplift processes- repetition of the canopy volcano area- what is that?? that lies above a stagnation chamber that has been carried out for this research project through geophysical investigation- confusing, rewrite this sentence."

Reply: The sentences were rewritten and clarified (Lines 83-85)

Methods Line 98: Figure 3 Incomplete Legend -what are the other colors and symbols in the map? -probably choose different background map. -what is the CRS /grid?

Reply: The legend has been maintained. The base map has been changed (Line 110).

Line 117-119: "During the monitoring sessions there has been ongoing research on the best Ground Control Points (GCPs) methods of acquisition. This was aimed at a more detailed indication of the 3D deformation process of the volcanic cap and a vertical and horizontal geometrical resolution of centimetre/subcentimetric order of magnitude." – confusing. Rewrite those sentences.

Reply: The sentence has been clarified and rewritten (Line 129-130).

Line 116 and onwards: Ground Control Points (GCPs) Please explain the different errors you mention and how they are calculated. Explain "error", "total error", "GCP error" and "average total 3D error" As well clarify for the reader what the difference between "GCPs", "Control Points" and "Checkpoints" are.

Reply: The error is calculated by Photoscan, through the mean square deviation of the distances of the CheckPoints with the point cloud. A description of the different types of errors has been added (Line 131-134). A definition of GCPs, Control Points and Check Points has been added (Line 171-176).

Line 122: "10-1:10-2 m"- please clarify what this means

Reply: Typing error has been corrected (Line 132)

Line 127-128: confusing, statistical analysis of what? The whole sentence should be clarified.

Reply: We meant: statistical error analysis. The sentence has been clarified and rewritten. (Line 142-144)

Line 129-130: "Since 2019 only TST has been used for the GCP survey and according to Tahar et al. (2013) the number of GCPs has been increased (Tab.1) reducing the

error to ≈ 1.4 cm and in the last campaign ≈ 0.7 cm." -are you saying the increase of the GCPs has reduced the GCP error or the fact that only TST was used for the survey? please clarify.

Reply: The joint application of more GCPs with the use of TST reduces the error (Line 144-146).

Line 131: ". . .greater control.."- what do you mean by that? – higher accuracy?

Reply: We meant more accurate geometry reconstruction of the models produced by SfM. The sentence was deleted.

Line 131:". . . limit number. . ."- explain what you mean by this.

Reply: We meant: increasing the number of GCPs. The sentence was deleted.

Figure 4: it would be beneficial to have a relative scale of the horizontal error below the green ellipse e.g.: (in X- direction) or something similar.

Reply: In figure 4 the scale has been improved as suggested

Line 151-152: "Photoscan accuracy" – what do you mean? The accuracy of the resulting data? Clarify

Reply: "Photoscan accuracy" is the term used by the Photoscan software to refer to instrument precision. This sentence has been rewritten and clarified (Line 166).

Figure 6: what do red and blue mean?

Reply: Red are the Control Points. Blue are the Checkpoints Control Points. The figure has been moved on figure 12.

Line 164-165: "25âŮę to 75âŮę percentile" – do you mean 25th to 75th percentile?

Reply: Of course, it was a typing error (Lines 286-287).

Photo Acquisition Line 173-174: "The camera was oriented in a 90- degree angle"- can

be confusing for the reader. Say that "vertical imagery" was acquired

Reply: We agree, the sentence has been rewritten (Line 186-187).

Line 174: explain what a single grid is.

Reply: We have added the explanation of single grid (Line 187-188).

Line 174: Introduce the acronym (ground sampling distance)

Reply: We have added the explanation of single grid (Line 188).

Data Processing SfM

Line 189-190: "value" doesn't seem the right term here. Rather use step/s of the processing chain

Reply: we followed the suggestion (Line 192).

Line 203: "measurements of check" -checkpoint measurement??

Reply: We meant measures applied to each image and checkpoint (Line 212-213).

Line 215: which version of Cloud Compare?

Reply: The version is v. 2.11 (Line 231).

Line 219-220: "This methodology has been chosen having in mind the heterogeneous distribution of the points in the sparse cloud, avoiding holes and thus null values." - is this why you chose this interpolation method or is it just something you keep in mind? - clarify and replace the term "having in mind" with something more precise. eg.: "This method has been chosen "because of"/ "due to" the heterogeneous distribution...."

Reply: - Yes, that is why this method of interpolation was chosen. We followed the suggestion by rewriting the sentence (Line 235-236).

Line 221: which "measurements" do you mean?

Reply: we meant comparison surveys (Line 244).

Line 223-224: it would be beneficial to include an illustration of a precision map.

Reply: Illustrations of precision maps and precision maps interpolated with the dense point cloud have been added (Line 222-227 and line 235 and line 238-241).

Figure 9: Numbers on legends too small

Reply: The scale numbers have been enlarged and the figure has been moved to figure 10.

Figure 11 and Line 264-265: "These were used to have an instrumental sensitivity scale of the measures of figure 12. The first and lowest one (âĹij2 cm) was easily detected." - what about the others? up to 10 cm? or are you saying that if only one is easily detected the others are not necessary. If you are not going to talk about the other calipers it should be considered whether figure 11 is necessary.

Reply: All calipers have been recorded but we have put the smallest as an example on the text. Fig.11 has been deleted following your suggestion (Line 265-268).

Discussion Line 289: "....temporally distant peaks...." – clarify what you mean by this? Data sets?

Reply: We meant: the greater temporal distance of the data. The sentence has been rewritten (Line 314-315).

Line 299: clarify in the caption which M3C2 distance is shown here (between which datasets) - explain what processes lead to the M3C2 distance around the edges of the survey area? or is this due to lower precision and/or accuracy in these areas?

Reply: - The data used has been explained more clearly. We assume that the increase in distance at the edges is mostly due to material deposition (Line 321-323).

Figure 15 and 16: add error bars to the time series. Increase the size of the font on the

scale

Reply: Error bars have been added. The font has been enlarged. The figures have been renumbered to 16 and 17 (Line 338 and 351).

Conclusion

Line 330-341: this paragraph and references should be part of the introduction.

Reply: We agree, the paragraph has been moved to Introduction (Line 25-37).

Line 351: clarify what kind of interval you mean by "between 30 and 60 g"

Reply: We meant an interval of 30 to 60 days to carry out a new survey. The sentence has been rewritten and clarified (Line 407).

Line 352-353: What does this mean for other projects? Is this amount of GCPs scalable for different area sizes?

Reply: This study can be useful for other projects, not in terms of the number of GCPs but in terms of methodology.

3. Technical Comments

Here are I point out minor technical suggestions with the same color coding as before.

Line 1: write "monitoring" with lower case m

Reply: Has been changed to lower case (Line 1).

Line 2: "the case study of "a"/"the" mud volcano

Reply: "The case study of the mud volcano" (Line 2).

Line 16: replace "danger" with "hazards"

Reply: Has been changed (Line 14).

Line 18: "for monitoring of deformation processes. . ."

Reply: Has been added (Line 18).

Line 21: Introduce abbreviation SfM

Reply: the abbreviation has been added (Line 20).

Line 34: "The different acquisition methods. . .."

Reply: It has been deleted (Line 47).

Line 35 - 36: "In this context, UAVs therefore offer unprecedented spatial and temporal resolution. . .." - repetition of the previous sentence.

Reply: The sentence was rewritten (Line 47- 48).

Line 38: ". . ...Lidar, thermal imaging cameras,. . ..." – add the comma

Reply: Comma was added (Line 50)

Line 44: ". . . is also affected by other features, for example. . .." –insert: controlled by other variables, such as:. . ..

Reply: The suggestion has been followed and the sentence has been rewritten (Line 55).

Line 44 -45: "design and altitude of the flight" flight path and flight altitude

Reply: The suggestion has been followed and the sentence has been rewritten (Line 56-57).

Line 48- 49: 95% (LoD 95%) Level of Detection - 95% Level of Detection (LoD 95%)

Reply: The sentence has been rewritten (Line 61-62).

Line 55: "Ortho-photos generated by UAVs in the area of the Santa Barbara mud volcano."- Ortho-photo of the Santa Barbara mud volcano generated from UAV imagery

Reply: The sentence has been deleted and the figure has been changed (Line 67).

Line 58: ". . ..developed since Late Miocene untill the Quaternary" – ". . .which developed from the Late Miocene to the Quaternary,. . ..

Reply: The sentence has been rewritten following the suggestion (Line 70).

Line 59: ". . ...formed by a foreland fold and a thrust belt. . .."

Reply: Indefinite articles have been added (Line 71-72).

Line 60: ". . ..the clastic sediments deposited. . .."- ". . .the deposition of clastic sediments. . ."

Reply: The sentence has been rewritten following the suggestion (Line 73).

Line 60-61: ". . ..during the late Miocene towards the Pleistocene. . .."- from the late Miocene to the Pleistocene

Reply: The sentence has been rewritten following the suggestion (Line 73-74).

Line 65: "On the mud volcanoes. . ."- which mud volcanoes??

Reply: Examples of mud volcanoes have been added (Line 77).

Line 62: "More hover. . .." ?

Reply: Typing error (Line 80).

Line 73. Personal communication with who? Which methods where used?

Reply: The personal communication was given by a colleague, he was quoted (Imposa et al., 2018), they applied tomography techniques (Line 85-86).

Line 73: "sill-like"

Reply: The word has been replaced (Line 86).

Line 74: replace develops with "sits" or "is located"

Reply: The word has been replaced with "is located" (Line 85).

Line 74: 50 meters

Reply: The word has been added but with "m" (Line 86).

Line 81: "On the surface outcrops deformation structures . . ." – rewrite

Reply: The sentence has been rewritten (Line 94).

Line 82-83:". . . highlighting what high intensity of stress and strain the volcano can generate. . ." ... highlighting the high stress and strain environment .....

Reply: The sentence has been rewritten (Line 95)

Line 84-85:". . ..and we still believe they are active. . .." .... and are still believed to be active,

Reply: The sentence has been rewritten (Line 96).

Line 91: monitoring – monitor

Reply: The word has been replaced by monitor (Line 103).

Line 113: introduce acronym for TST

Reply: The acronym has been introduced (Line 106).

Line 113: ". . .TST base local station. . ." – local base station / local TST base station???

Reply: The sentence has been rewritten (Line 106-107).

Line 120: Real Time Kinematics (RTK)

Reply: The acronym has been added (Line 130).

Line 121: GCPs

Reply: Has been corrected (Line 131).

Line 121: ". . .(Tab. 2).." – do you mean Table 1??

Reply: Yes I do (Line 132-133)

Line 125: "detected"-"measured"

Reply: The word has been changed by "measured" (Line 141).

Table 1: "GCPs NUMBER" – GCP NUMBER

Reply: We followed this suggestion

Line 135: ". . .on the table. . ." – in the table

Reply: The caption has been rewritten (Line 149).

Line 141: rephrase the sentence.

Reply: The sentence has been rephrasing (Line 156-157).

Line 152: ". . .CTN1 points coordinates. . ."

Reply: The correction has been performed (Line 178).

Line 175: ". . . a cheap UAV." – redundant information. Rather mention that it is a quadcopter UAV

Reply: The suggestion was implemented (Line 183).

Line 175: ". . .. Fly flight altitude. . .. . ."

Reply: The word has been replaced (Line 183).

Line 176. "The feature of. . .."- feature is the wrong term here. Just say: The sensor size is. . ...

Reply: The suggestion was implemented (Line 184).

Line 193: add "" to the Gradual Selection

Reply: The quotation marks has been added (Line 198).

Line 196-197: James, et al. . .. . ..

Reply: Has been corrected (Line 218).

Line 210: "Secondly,. . ." where is the firstly??

Reply: There is no firstly, the sentence has been rewritten (Line 219).

Line 248: significant change

Reply: Letters have been replaced (Line 304-305).

Line 275: ". . .survey techniques, they have a very. . .."

Reply: The sentences has been rephrased (Line 289).

Line 289: "We....."

Reply: The sentence has been rewritten (Line 316-318).

Line 290: "Data are still. . ..."- "The data is still . . ..

Reply: The sentence has been rewritten (Line 316-317).

Line 294: "In literature. . ." – "According to literature. . .."

Reply: The sentence has been rewritten (Line 395-396).

Line 360: "In this case more thoroughly monitoring will carried out:. . ."- rewrite this sentence.

Reply: The sentence has been rewritten (Line 416-418)

Please also note the supplement to this comment:
https://nhess.copernicus.org/preprints/nhess-2020-378/nhess-2020-378-SC1-supplement.pdf

**Supplement:**

[revised manuscript text omitted]

---

## Short Comment (SC2) · Fabio Brighenti et al. · 10 Feb 2021

Reply to Reviewer 2 RC2

UAV survey method to Monitoring and analysing geological hazards: The Case study of mud volcano of Villaggio Santa Barbara, Caltanissetta (Sicily)

Fabio Brighenti et al.

A pdf file of the reviewed manuscript is attached as supplement file. The changes in

the text are marked with green for rephrasing, reorganised and new parts added, and with yellow the changes suggested by the referees.

The list of references in the introduction section must be updated with recentmost researches (2018-2020), they are many.

Reply: We have followed the suggestion. The bibliography has been improved, including papers from 2018 to 2021.

The text regarding the GCP in the section must be moved to the methods.

Reply: The text concerning GCPs has been moved to the methods chapter.

Figure 1 show the orthomosaic resulting from photogrammetry processing. It can be moved to the results section. A geological introduction to the area is mandatory.

Reply: We have substituted Figure 1 with an aerial photo of the study area. The geological introduction is present in the text.

Report the reference system in all figures.

Reply: The reference system has been added to all figures.

There is a lack of references in the methods as well as there is a lack of details regarding the processing. I suggest reading recentmost papers focusing on the methodology.

Reply: The chapter of methodology was improved and clarified through rephrasing and the addition of the Results chapter. The bibliography has been enhanced.

It has been generated a DSM (Digital surface model) not a DEM.

Reply: The acronym DEM is not present in the paper.

Conclusion must be rewritten removing the lines listing several references that can be added to the introduction section. Conclusions must address the main results from the present work.

Reply: The conclusion has been rewritten. The paragraph with the long list of references has been moved to the Introduction chapter.

Please also note the supplement to this comment:
https://nhess.copernicus.org/preprints/nhess-2020-378/nhess-2020-378-SC2-supplement.pdf

**Supplement:**

[revised manuscript text omitted]

---

## Short Comment (SC3) · Fabio Brighenti et al. · 10 Feb 2021

UAV survey method to Monitoring and analysing geological hazards: The Case study of mud volcano of Villaggio Santa Barbara, Caltanissetta (Sicily)

Fabio Brighenti et al.

A pdf file of the reviewed manuscript is attached as supplement file. The changes in the text are marked with green for rephrasing, reorganised and new parts added, and with yellow the changes suggested by the referees.

[Figure]

I suggest an improvement for the literature, some recent and important paper are missing.

Reply: We have followed the suggestion. The bibliography has been improved, including papers from 2018 to 2020.

The methodology must be improved, both regarding the UAV survey and SfM processing.

Reply: The chapter of methodology was improved and clarified through rephrasing and the addition of the Results chapter.

Figure 1 cannot be the Orthomosaic resulting from processing, it is a result

Reply: We have substituted Figure 1 with an aerial photo of the study area.

Conclusions must be rewritten avoiding the long list of citation.

Reply: The conclusion has been rewritten. The paragraph with the long list of references has been moved to the Introduction chapter.

Add the reference system in all figures.

Reply: The reference system has been added to all figures.

Please also note the supplement to this comment:
https://nhess.copernicus.org/preprints/nhess-2020-378/nhess-2020-378-SC3-supplement.pdf

---

## Author Comment (AC1) · 11 Feb 2021

dear reviewer, I thank yourself for very helpful comments, nearly all of which have been followed in revising the manuscript. More specifically, with the coauthors we discussed your comments and answered all your suggestions. The details of the changes are listed in the enclosed by dr Fabio Brighenti who is the second contact and first author of the manuscript.

———————————————————

2020-378, 2020.

---

## Author Response (AR1)

**Reply to Reviewer 1**

UAV survey method to Monitoring and analysing geological hazards: The Case study of mud volcano of Villaggio Santa Barbara, Caltanissetta (Sicily)

Fabio Brighenti et al.

A pdf file of the reviewed manuscript is attached as supplement file. The changes in the text are marked with green for rephrasing, reorganised and new parts added, and with yellow the changes suggested by the referees.

**1. General Comments**

Hereby I want to state that I reviewed this manuscript to my best ability and approached it with interest and with the intention to give suggestions which improve the quality of the work. The methods used in the paper are interesting and relevant for future studies in order to optimize SfM methods for ground deformation analysis and monitoring. However, the paper needs reorganization, clarification and rephrasing. I want to address that the presented manuscript does not contain a Results chapter but instead goes directly from Methods to Discussion. However, the results of the study are presented in both Methods and Discussion. This makes the article confusing and resulting data unclear. I would strongly recommend a reorganization of the content with a clearer separation between Methods, Results and Discussion. Doing this will increase the readability of the work significantly. Furthermore, the first paragraph of the Conclusion would make more sense in the Introduction chapter. I would as well strongly recommend having this paper proof-read by a native English speaker before the next submission to correct and improve the phrasing. The sentence structure is sometimes too complicated or incorrect so that I think the paper quality would largely benefit from having it proof-read.

> *Reply: Thank you for your interest in understanding and helping us with the paper. We want to thank you for your kind advices. The paper has been reorganised; the chapter Results has been added. The paper now follows the order of the chapters Methods, Results and Discussion. The data has been reordered within the Results chapter. Large parts of the paper have been reorganised, clarified and rephrased. The first paragraph of the Discussion has been moved to the Introducion. The paper has been corrected by a native speaker.*

**2. Specific Comments**

**Abstract**

Line 12- 15: "Among all the active geological processes on Santa Barbara mud volcano (Caltanissetta town, Italy), represents a dangerous site because it caused, on 11 August 2008, a paroxysmal event, which determined severe damages to the infrastructures at around to 2 km the paroxysmal event." Confusing - rephrase this sentence.

> *Reply: The sentence has been rewritten (Line 15-17).*

**Introduction**

Line 40 – 44: "The accuracy of 3D information can be significantly increased by the ground control points (GCPs) by georeferencing the data to ground control points (GCPs). The GCPs acquisition acquisition is a undamental fundamental aspect of the georeferencing of the network of images captured by UAV photogrammetry – unclear phrasing. Images are not captured by UAV photogrammetry. Images are captured "for photogrammetry" or "by UAV". "In this process a right number of GCPs is required which lead to a greater accuracy of the outcomes (point clouds, 3D grid, orthomosaic or digital surface model (DSM))." - rewrite these sentences more concise without repetition. e.g.: "Ground control points (GCPs) are used to improve the accuracy of the resulting data. Therefore, points recognizable on the UAV imagery are measured with a survey device to georeference the data."

*Reply: The sentences were rewritten and clarified (Lines 53-54).*

Line 46: "In this paperthrough SFM,. . ."- what do you mean? As well use a consistent acronym for SfM

*Reply: There was a typo in the sentence. The acronym is inserted (Line 20).*

Line 70 – 73: "This morphometric structure is typical of uplifting areas and therefore relative decrease of the base level. This morphometric evidence suggests uplift processes- repetition of the canopy volcano area- what is that?? that lies above a stagnation chamber that has been carried out for this research project through geophysical investigation- confusing, rewrite this sentence."

*Reply: The sentences were rewritten and clarified (Lines 83-85)*

**Methods**

Line 98: Figure 3 Incomplete Legend -what are the other colors and symbols in the map? -probably choose different background map. -what is the CRS /grid?

*Reply: The legend has been maintained. The base map has been changed (Line 110).*

Line 117-119: "During the monitoring sessions there has been ongoing research on the best Ground Control Points (GCPs) methods of acquisition. This was aimed at a more detailed indication of the 3D deformation process of the volcanic cap and a vertical and horizontal geometrical resolution of centimetre/subcentimetric order of magnitude." – confusing. Rewrite those sentences.

*Reply: The sentence has been clarified and rewritten (Line 129-130).*

Line 116 and onwards: Ground Control Points (GCPs) Please explain the different errors you mention and how they are calculated. Explain "error", "total error", "GCP error" and "average total 3D error" As well clarify for the reader what the difference between "GCPs", "Control Points" and "Checkpoints" are.

*Reply: The error is calculated by Photoscan, through the mean square deviation of the distances of the CheckPoints with the point cloud. A description of the*

*different types of errors has been added (Line 131-134). A definition of GCPs, Control Points and Check Points has been added (Line 171-176).*

Line 122: "10-1:10-2 m"- please clarify what this means

*Reply: Typing error has been corrected (Line 132)*

Line 127-128: confusing, statistical analysis of what? The whole sentence should be clarified.

*Reply: We meant: statistical error analysis. The sentence has been clarified and rewritten. (Line 142-144)*

Line 129-130: "Since 2019 only TST has been used for the GCP survey and according to Tahar et al. (2013) the number of GCPs has been increased (Tab.1) reducing the error to ≈ 1.4 cm and in the last campaign ≈ 0.7 cm." -are you saying the increase of the GCPs has reduced the GCP error or the fact that only TST was used for the survey? please clarify.

*Reply: The joint application of more GCPs with the use of TST reduces the error (Line 144-146).*

Line 131: ". . .greater control.."- what do you mean by that? – higher accuracy?

*Reply: We meant more accurate geometry reconstruction of the models produced by SfM. The sentence was deleted.*

Line 131:". . . limit number. . ."- explain what you mean by this.

*Reply: We meant: increasing the number of GCPs. The sentence was deleted.*

Figure 4: it would be beneficial to have a relative scale of the horizontal error below the green ellipse e.g.: (in X- direction) or something similar.

*Reply: In figure 4 the scale has been improved as suggested*

Line 151-152: "Photoscan accuracy" – what do you mean? The accuracy of the resulting data? Clarify

*Reply: "Photoscan accuracy" is the term used by the Photoscan software to refer to instrument precision. This sentence has been rewritten and clarified (Line 166).*

Figure 6: what do red and blue mean?

*Reply: Red are the Control Points. Blue are the Checkpoints Control Points. The figure has been moved on figure 12.*

Line 164-165: "25◦ to 75◦ percentile" – do you mean 25th to 75th percentile?

*Reply: Of course, it was a typing error (Lines 286-287).*

**Photo Acquisition**

Line 173-174: "The camera was oriented in a 90- degree angle"- can be confusing for the reader. Say that "vertical imagery" was acquired

> *Reply: We agree, the sentence has been rewritten (Line 186-187).*

Line 174: explain what a single grid is.

> *Reply: We have added the explanation of single grid (Line 187-188).*

Line 174: Introduce the acronym (ground sampling distance)

> *Reply: We have added the explanation of single grid (Line 188).*

**Data Processing SfM**

Line 189-190: "value" doesn't seem the right term here. Rather use step/s of the processing chain

> *Reply: we followed the suggestion (Line 192).*

Line 203: "measurements of check" -checkpoint measurement??

> *Reply: We meant measures applied to each image and checkpoint (Line 212-213).*

Line 215: which version of Cloud Compare?

> *Reply: The version is v. 2.11 (Line 231).*

Line 219-220: "This methodology has been chosen having in mind the heterogeneous distribution of the points in the sparse cloud, avoiding holes and thus null values." - is this why you chose this interpolation method or is it just something you keep in mind? - clarify and replace the term "having in mind" with something more precise. eg.: "This method has been chosen "because of"/ "due to" the heterogeneous distribution...."

> *Reply: - Yes, that is why this method of interpolation was chosen. We followed the suggestion by rewriting the sentence (Line 235-236).*

Line 221: which "measurements" do you mean?

> *Reply: we meant comparison surveys (Line 244).*

Line 223-224: it would be beneficial to include an illustration of a precision map.

> *Reply: Illustrations of precision maps and precision maps interpolated with the dense point cloud have been added (Line 222-227 and line 235 and line 238-241).*

Figure 9: Numbers on legends too small

> *Reply: The scale numbers have been enlarged and the figure has been moved to figure 10.*

Figure 11 and Line 264-265: "These were used to have an instrumental sensitivity scale of the measures of figure 12. The first and lowest one (~2 cm) was easily detected." - what about the others? up to 10 cm? or are you saying that if only one is easily detected the others are not necessary. If you are not going to talk about the other calipers it should be considered whether figure 11 is necessary.

> *Reply: All calipers have been recorded but we have put the smallest as an example on the text. Fig.11 has been deleted following your suggestion (Line 265-268).*

**Discussion**

Line 289: "....temporally distant peaks...." – clarify what you mean by this? Data sets?

> *Reply: We meant: the greater temporal distance of the data. The sentence has been rewritten (Line 314-315).*

Line 299: clarify in the caption which M3C2 distance is shown here (between which datasets) - explain what processes lead to the M3C2 distance around the edges of the survey area? or is this due to lower precision and/or accuracy in these areas?

> *Reply: - The data used has been explained more clearly. We assume that the increase in distance at the edges is mostly due to material deposition (Line 321-323).*

Figure 15 and 16: add error bars to the time series. Increase the size of the font on the scale

> *Reply: Error bars have been added. The font has been enlarged. The figures have been renumbered to 16 and 17 (Line 338 and 351).*

**Conclusion**

Line 330-341: this paragraph and references should be part of the introduction.

> *Reply: We agree, the paragraph has been moved to Introduction (Line 25-37).*

Line 351: clarify what kind of interval you mean by "between 30 and 60 g"

> *Reply: We meant an interval of 30 to 60 days to carry out a new survey. The sentence has been rewritten and clarified (Line 407).*

Line 352-353: What does this mean for other projects? Is this amount of GCPs scalable for different area sizes?

*Reply: This study can be useful for other projects, not in terms of the number of GCPs but in terms of methodology.*

**3. Technical Comments**

Here are I point out minor technical suggestions with the same color coding as before.

Line 1: write "monitoring" with lower case m

> *Reply: Has been changed to lower case (Line 1).*

Line 2: "the case study of "a"/"the" mud volcano

> *Reply: "The case study of the mud volcano" (Line 2).*

Line 16: replace "danger" with "hazards"

> *Reply: Has been changed (Line 14).*

Line 18: "for monitoring of deformation processes. . ."

> *Reply: Has been added (Line 18).*

Line 21: Introduce abbreviation SfM

> *Reply: the abbreviation has been added (Line 20).*

Line 34: "The different acquisition methods. . .."

> *Reply: It has been deleted (Line 47).*

Line 35 - 36: "In this context, UAVs therefore offer unprecedented spatial and temporal resolution. . .." - repetition of the previous sentence.

> *Reply: The sentence was rewritten (Line 47- 48).*

Line 38: ". . ...Lidar, thermal imaging cameras,. . ..." – add the comma

> *Reply: Comma was added (Line 50)*

Line 44: ". . . is also affected by other features, for example. . .." –insert: controlled by other variables, such as:. . ..

> *Reply: The suggestion has been followed and the sentence has been rewritten (Line 55).*

Line 44 -45: "design and altitude of the flight" flight path and flight altitude

*Reply: The suggestion has been followed and the sentence has been rewritten (Line 56-57).*

Line 48- 49: 95% (LoD 95%) Level of Detection - 95% Level of Detection (LoD 95%)

*Reply: The sentence has been rewritten (Line 61-62).*

Line 55: "Ortho-photos generated by UAVs in the area of the Santa Barbara mud volcano."- Ortho-photo of the Santa Barbara mud volcano generated from UAV imagery

*Reply: The sentence has been deleted and the figure has been changed (Line 67).*

Line 58: ". . ..developed since Late Miocene untill the Quaternary" – ". . .which developed from the Late Miocene to the Quaternary,. . ..

*Reply: The sentence has been rewritten following the suggestion (Line 70).*

Line 59: ". . ...formed by a foreland fold and a thrust belt. . .."

*Reply: Indefinite articles have been added (Line 71-72).*

Line 60: ". . ..the clastic sediments deposited. . .."- ". . .the deposition of clastic sediments. . ."

*Reply: The sentence has been rewritten following the suggestion (Line 73).*

Line 60-61: ". . ..during the late Miocene towards the Pleistocene. . .."- from the late Miocene to the Pleistocene

*Reply: The sentence has been rewritten following the suggestion (Line 73-74).*

Line 65: "On the mud volcanoes. . ."- which mud volcanoes??

*Reply: Examples of mud volcanoes have been added (Line 77).*

Line 62: "More hover. . .."  ?

*Reply: Typing error (Line 80).*

Line 73. Personal communication with who? Which methods where used?

*Reply: The personal communication was given by a colleague, he was quoted (Imposa et al., 2018), they applied tomography techniques (Line 85-86).*

Line 73: "sill-like"

*Reply: The word has been replaced (Line 86).*

Line 74: replace develops with "sits" or "is located"

*Reply: The word has been replaced with "is located" (Line 85).*

Line 74: 50 meters

*Reply: The word has been added but with "m" (Line 86).*

Line 81: "On the surface outcrops deformation structures . . ." – rewrite

*Reply: The sentence has been rewritten (Line 94).*

Line 82-83:". . . highlighting what high intensity of stress and strain the volcano can generate. . ." … highlighting the high stress and strain environment .....

*Reply: The sentence has been rewritten (Line 95)*

Line 84-85:". . ..and we still believe they are active. . .." .... and are still believed to be active,

*Reply: The sentence has been rewritten (Line 96).*

Line 91: monitoring – monitor

*Reply: The word has been replaced by monitor (Line 103).*

Line 113: introduce acronym for TST

*Reply: The acronym has been introduced (Line 106).*

Line 113: ". . .TST base local station. . ." – local base station / local TST base station???

*Reply: The sentence has been rewritten (Line 106-107).*

Line 120: Real Time Kinematics (RTK)

*Reply: The acronym has been added (Line 130).*

Line 121: GCPs

*Reply: Has been corrected (Line 131).*

Line 121: ". . .(Tab. 2).." – do you mean Table 1??

*Reply: Yes I do (Line 132-133)*

Line 125: "detected"-"measured"

*Reply: The word has been changed by "measured" (Line 141).*

Table 1: "GCPs NUMBER" – GCP NUMBER

*Reply: We followed this suggestion*

Line 135: ". . .on the table. . ." – in the table

*Reply: The caption has been rewritten (Line 149).*

Line 141: rephrase the sentence.

*Reply: The sentence has been rephrasing (Line 156-157).*

Line 152: ". . .CTN1 points coordinates. . ."

*Reply: The correction has been performed (Line 178).*

Line 175: ". . . a cheap UAV.." – redundant information. Rather mention that it is a quadcopter UAV

*Reply: The suggestion was implemented (Line 183).*

Line 175: ". . .. Fly flight altitude. . .. . ."

*Reply: The word has been replaced (Line 183).*

Line 176. "The feature of. . .."- feature is the wrong term here. Just say: The sensor size is. . ...

*Reply: The suggestion was implemented (Line 184).*

Line 193: add "" to the Gradual Selection

*Reply: The quotation marks has been added (Line 198).*

Line 196-197: James, et al. . .. . ..

*Reply: Has been corrected (Line 218).*

Line 210: "Secondly,. . ." where is the firstly??

*Reply: There is no firstly, the sentence has been rewritten (Line 219).*

Line 248: significant change

*Reply: Letters have been replaced (Line 304-305).*

Line 275: ". . .survey techniques, they have a very. . .."

*Reply: The sentences has been rephrased (Line 289).*

Line 289: "We....."

*Reply: The sentence has been rewritten (Line 316-318).*

Line 290: "Data are still. . ..."- "The data is still . . ..

*Reply: The sentence has been rewritten (Line 316-317).*

Line 294:  "In literature. . ." – "According to literature. . .."

*Reply: The sentence has been rewritten (Line 395-396).*

Line 360: "In this case more thoroughly monitoring will carried out:. . ."- rewrite this sentence.

*Reply: The sentence has been rewritten (Line 416-418).*

**Reply to Reviewer 2 RC2**

UAV survey method to Monitoring and analysing geological hazards: The Case study of mud volcano of Villaggio Santa Barbara, Caltanissetta (Sicily)

Fabio Brighenti et al.

A pdf file of the reviewed manuscript is attached as supplement file. The changes in the text are marked with green for rephrasing, reorganised and new parts added, and with yellow the changes suggested by the referees.

The list of references in the introduction section must be updated with recentmost researches (2018-2020), they are many.

>*Reply: We have followed the suggestion. The bibliography has been improved, including papers from 2018 to 2021.*

The text regarding the GCP in the section must be moved to the methods.

>*Reply: The text concerning GCPs has been moved to the methods chapter.*

Figure 1 show the orthomosaic resulting from photogrammetry processing. It can be moved to the results section. A geological introduction to the area is mandatory.

>*Reply: We have substituted Figure 1 with an aerial photo of the study area. The geological introduction is present in the text.*

Report the reference system in all figures.

>*Reply: The reference system has been added to all figures.*

There is a lack of references in the methods as well as there is a lack of details regarding the processing. I suggest reading recentmost papers focusing on the methodology.

>*Reply: The chapter of methodology was improved and clarified through rephrasing and the addition of the Results chapter. The bibliography has been enhanced.*

It has been generated a DSM (Digital surface model) not a DEM.

>*Reply: The acronym DEM is not present in the paper.*

Conclusion must be rewritten removing the lines listing several references that can be added to the introduction section. Conclusions must address the main results from the present work.

>*Reply: The conclusion has been rewritten. The paragraph with the long list of references has been moved to the Introduction chapter.*

**Reply to Reviewer 2 RC3**

UAV survey method to Monitoring and analysing geological hazards: The Case study of mud volcano of Villaggio Santa Barbara, Caltanissetta (Sicily)

Fabio Brighenti et al.

A pdf file of the reviewed manuscript is attached as supplement file. The changes in the text are marked with green for rephrasing, reorganised and new parts added, and with yellow the changes suggested by the referees.

I suggest an improvement for the literature, some recent and important paper are missing.

*Reply: We have followed the suggestion. The bibliography has been improved, including papers from 2018 to 2020.*

The methodology must be improved, both regarding the UAV survey and SfM processing.

*Reply: The chapter of methodology was improved and clarified through rephrasing and the addition of the Results chapter.*

Figure 1 cannot be the Orthomosaic resulting from processing, it is a result

*Reply: We have substituted Figure 1 with an aerial photo of the study area.*

Conclusions must be rewritten avoiding the long list of citation.

*Reply: The conclusion has been rewritten. The paragraph with the long list of references has been moved to the Introduction chapter.*

Add the reference system in all figures.

*Reply: The reference system has been added to all figures.*

---

## Referee Report (RR1)

Review

General comments:
The manuscript has significantly improved. Very interesting read. I only have few minor comments.

Specific comments:

Line 12: "Nowadays  remote sensing…."

Line 13:"…. and monitoring of  natural hazards."

Line 47: " … spatial and temporal resolution"

Figure 2 & Figure 3: *Just a suggestion:* it would be nice to combine the two figures into one.

Line 197: "….but we modified  some parameters during the cleaning procedure of the sparse point cloud."

Line 60/61, 254 and 262: Unify the formatting of LoD percentage font size

Line 270: "On the upper part of the figure the dense point clouds of the two surveys of the same area carried out on 2019/07/29 (left)….."

Line 277: Rewrite this sentence. Here is a suggestion: "As illustrated  in  section 2.2, the results show that using  between 40 and 60 % of the GCPs  as control points)  the RMSE value has  minimal variation."

376: "After  the uncertainty and sensitivity of the surveys were computed, the time series were made."

---

## Referee Report (RR2)

Change title to 'UAV survey method to monitor and analyze geological hazards: ….. '

Line 13: Change in 'of geological hazards'.

Line 14: Remove 'and monitoring'.

Line 20: Structure from Motion (SfM). Correct this also in the other parts of the manuscript.

Line 25: Softwares.

Lines 27-30: Citations should be ordered by year of publication.

Line 35: Add papers from Bonali et al. which are based on Structure from Motion techniques and drone surveys applied to active volcanic areas.

Line 43: Remove 'in'.

Line 47: What do you mean by high 'temporal resolutions'? Please detail better.

Line: 'to georeference'.

Line 55: in other parts of the manuscript, you speak about DEMs, not DSMs. Please make the text more homogeneous.

Line 72: Fold and thrust belt. Correct also in 'gradually deformed'. Remove 'and moved'.

Line 74: they 'represent' a preferential way.

Line 94: 'outside the volcano area'.

Line 96: Why are you supposing that these structures are active? Please specify.

Line 104: 'in particular ensuring i)…..'

Line 119: '0.016 m'.

Line 121: 'to compute'.

Line 123: '0.024 m'.

Line 131: 'GCPs'.

Line 142: The sentence is not clear, please rephrase.

Line 144: Please rephrase.

Line 156: Please specify the acronym 'TST'.

Line 158: Remove the comma after 'network'. Also 'Fig. 5A'. Same in line 161 '(Fig. 5B)'.

Line 161: What do you mean by 'verticality'? What is a bubble level?

Line 166: The sentence is not clear, please rephrase.

Line 170: Change 'we have done' in 'we performed'. Also, what do you mean by 'compared with the SfM software data'?

Line 175: Remove 'during'.

Line 186: 'Nadir'.

Line 190: '2.4 Data processing through Structure from Motion (SfM) techniques'.

Line 192: 'Steps of the processing chain regarding Tie Point Accuracy and Marker Accuracy have been performed according to James et al. (2017) (Fig.6)'.

Line 196: we modified 'some' parameters.

Lines 195-200: These sentences can be improved.

Line 206: Remove the comma after James.

Line 229: is usually detected from 'sparse point clouds'.

Line 233: Dense point cloud. Also, 'is the most suitable'.

Line 235: Dense point clouds. Also in line 243.

Lines 250-252: Remove the space between 10 and A,B,C. Same in all text.

Line 253: 'which exceed'.

Line 267: 'of the measurements'.

Line 276: '3. Results'.

Line 277: 'As shown in section 2.2, using a percentage of GCPs…'.

Line 278: 'GCPs'.

Line 279: 'When the threshold of 60% is exceeded,…'

Line 315: 'on the whole mud volcano area'.

Line 316: 'In order to visualize important deformations….'

Line 345: 'Figure'. Same in line 346.

Line 360: Change 'questions' in 'issues'.

Line 366: 'the second aspect to be considered is the evaluation….'.

Line 371: What do you mean by 'volcano shell'?

Line 375: 'at least of…'.

Line 398: 'In hazards management,…'. Remove comma after 'technique'.

Line 403: 'can occur up to one year before…'.

Line 415: 'in the case of anomalous values detected, the monitoring campaigns must be improved'.

Figure 1: Add a scale and an orientation to the figure.

Figure 2: North and scale are missing.

Figure 3: Correct 'local' in the legend.

Figure 6: Correct 'Adjustment' and 'Reconstruction'. In Step 3, the white box is missing the word 'Point' in 'Build Dense Cloud'. Also, specify the acronym DEM.

Figure 17: in the legends, correct 'Area', sometimes written as 'areae'.

---

## Author Response (AR2)

**Reply to Reviewer 1**

UAV survey method to Monitoring and analysing geological hazards: The Case study of mud volcano of Villaggio Santa Barbara, Caltanissetta (Sicily)

Fabio Brighenti et al.

A pdf file of the reviewed manuscript is attached as supplement file. The changes in the text are marked with red for rephrasing, reorganized and the changes suggested by the referees.

**General comments:**

The manuscript has significantly improved. Very interesting read. I only have few minor comments.

*Reply: Thank you for reading our work with interest. We have improved the quality of the manuscript through the advice of all the reviewers. It was a great help in improving our manuscript.*

**Specific comments:**

Line 12: "Nowadays  remote sensing…."

*Reply: Word deleted.*

Line 13:"…. and monitoring of  natural hazards."

*Reply: The sentence has been changed.*

Line 47: " … spatial and temporal resolution"

*Reply: The letter has been deleted.*

Figure 2 & Figure 3: *Just a suggestion:* it would be nice to combine the two figures into one.

*Reply: Thanks for the suggestion but, we considered not to merge the two figures.*

Line 197: "….but we modified  some parameters during the cleaning procedure of the sparse point cloud."

*Reply: The word was changed, and the article has been added.*

Line 60/61, 254 and 262: Unify the formatting of LoD percentage font size

*Reply: The font size was unified.*

Line 270: "On the upper part of the figure the dense point clouds of the two surveys of the same area carried out on 2019/07/29 (left)....."

*Reply: The sentence has been corrected.*

Line 277: Rewrite this sentence. Here is a suggestion: "As illustrated  in  section 2.2, the results show that using  between 40 and 60 % of the GCPs  as control points the RMSE value has  minimal variation."

*Reply: The suggestion was followed; the sentence was rephrased.*

376: "After  the uncertainty and sensitivity of the surveys were computed, the time series were made."

*Reply: The sentence has been corrected.*

**Reply to Reviewer 1**

UAV survey method to Monitoring and analysing geological hazards: The Case study of mud volcano of Villaggio Santa Barbara, Caltanissetta (Sicily)

Fabio Brighenti et al.

A pdf file of the reviewed manuscript is attached as supplement file. The changes in the text are marked with red for rephrasing, reorganized and the changes suggested by the referees.

> *Reply: Thank you for your careful reading and suggestions, we think the discussion with you was helpful in improving the quality of the submitted paper.*

**Specific comments:**

Change title to 'UAV survey method to monitor and analyze geological hazards: ….. '

> *Reply: The title has been modified*

Line 13: Change in 'of geological hazards'.

> *Reply: We followed the first referee's suggestion.*

Line 14: Remove 'and monitoring'.

> *Reply: We have removed the part of the sentence.*

Line 20: Structure from Motion (SfM). Correct this also in the other parts of the manuscript.

> *Reply: We have checked the entire paper and corrected the typo.*

Line 25: Softwares.

> *Reply: The word has been corrected.*

Lines 27-30: Citations should be ordered by year of publication.

> *Reply: Citations have been sorted in chronological order*

Line 35: Add papers from Bonali et al. which are based on Structure from Motion techniques anddrone surveys applied to active volcanic areas.

> *Reply: Bibliography was improved with recommended author.*

Line 43: Remove 'in'.

> *Reply: It has been removed.*

Line 47: What do you mean by high 'temporal resolutions'? Please detail better.

> *Reply: By this phrase we meant data acquisition frequency in the order of magnitude of hours to a maximum of one day.*

Line 54: 'to georeference'.

> *Reply: The verbal tense was changed.*

Line 55: in other parts of the manuscript, you speak about DEMs, not DSMs. Please make the textmore homogeneous.

> *Reply: The term DEM has been replaced by DSM throughout the paper and in Figure 6.*

Line 72: Fold and thrust belt. Correct also in 'gradually deformed'. Remove 'and moved'.

> *Reply: The sentence has been reworded according to suggestions.*

Line 74: they 'represent' a preferential way.

> *Reply: The word has been changed.*

Line 94: 'outside the volcano area'.

> *Reply: The sentece has been modified.*

Line 96: Why are you supposing that these structures are active? Please specify.

> *Reply: We believe that the structures are active from the surveys carried out on 2002 and 2008, other evidence is the deformed athletics in the eastern part.*

Line 104: 'in particular ensuring i).....'

> *Reply: The sentence has been rearranged.*

Line 119: '0.016 m'.

> *Reply: Unit of measurement added.*

Line 121: 'to
compute'.

*Reply: The verb has been corrected.*

Line 123: '0.024
m'.

*Reply: Unit of measurement added.*

Line 131: 'GCPs'.

*Reply: The typing error has been
corrected.*

Line 142: The sentence is not clear, please
rephrase.

*Reply: The sentence has been
rephrased.*

Line 144: Please rephrase.

*Reply: The sentence has been rephrased.*

Line 156: Please specify the acronym 'TST'.

*Reply: The acronym has been reinserted.*

Line 158: Remove the comma after 'network'. Also 'Fig. 5A'. Same in line 161
'(Fig. 5B)'.

*Reply: Corrections have been made throughout the paper.*

Line 161: What do you mean by 'verticality'? What is a bubble level?

*Reply: The sentence has been rewritten, by "verticalty" we meant a measurement
perpendicular to GCPs. A spirit level, bubble level, or simply a level, is an instrument
designed to indicate whether a surface is horizontal (level) or vertical (plumb).*

Line 166: The sentence is not clear, please rephrase.

*Reply: The sentence has been rephrased.*

Line 170: Change 'we have done' in 'we performed'. Also, what do you mean by
'compared withthe SfM software data'?

*Reply: The verb has been changed. The sentence has been rewritten, explaining that
the comparison was with the previous results of PhotoScan.*

Line 175: Remove

'during'.

>    *Reply:*
Removed.

Line 186: 'Nadir'.

>    *Reply: The word has been corrected.*

Line 190: '2.4 Data processing through Structure from Motion (SfM) techniques'.

>    *Reply: The title of the paragraph has been changed.*

Line 192: 'Steps of the processing chain regarding Tie Point Accuracy and Marker Accuracy havebeen performed according to James et al. (2017) (Fig.6)'.

>    *Reply: The advice was followed.*

Line 196: we modified 'some' parameters.

>    *Reply: The word has been corrected.*

Lines 195-200: These sentences can be improved.

>    *Reply: Sentences have been*
>    *rephrased and improved*

Line 206: Remove the comma after James.

>    *Reply:Removed.*

Line 229: is usually detected from 'sparse point clouds'.

>    *Reply: The typo has been corrected.*

Line 233: Dense point cloud. Also, 'is the most suitable'.

>    *Reply: The suggestions were followed.*

Line 235: Dense point clouds. Also in line 243.

>    *Reply: The typo has been corrected.*

Lines 250-252: Remove the space between 10 and A,B,C. Same in all text.

*Reply: The space between the numbers and letters have all been eliminated.*

Line 253: 'which exceed'.

*Reply: Was correct.*

Line 267: 'of the measurements'.

*Reply: Has been corrected word*

Line 276: '3. Results'.

*Reply: has been corrected the title*

Line 277: 'As shown in section 2.2, using a percentage of GCPs…'.

*Reply: The sentence has been modified.*

Line 278: 'GCPs'.

*Reply: The typo has been corrected.*

Line 279: 'When the threshold of 60% is exceeded,

*Reply: The sentence has been modified.*

Line 315: 'on the whole mud volcano area'.

*Reply: The suggestion has been made in the text.*

Line 316: 'In order to visualize important deformations….'

*Reply: The sentence has been modified.*

Line 345: 'Figure'. Same in line 346.

*Reply: The capital letter has been added.*

Line 360: Change 'questions' in 'issues'.

*Reply: The word has been changed.*

Line 366: 'the second aspect to be considered is the evaluation….'.

*Reply: The sentence has been amended.*

Line 371: What do you mean by 'volcano shell'?

*Reply: The sentence has been changed. We meant the bare mud surface.*

Line 375: 'at least of…'.

*Reply: The advice was followed.*

Line 398: 'In hazards management,…'. Remove comma after 'technique'.

*Reply: The suggestion has been made in the text.*

Line 403: 'can occur up to one year before…'.

*Reply: Has been added in the paper.*

Line 415: 'in the case of anomalous values detected, the monitoring campaigns must be improved'.

*Reply: The sentence has been modified.*

Figure 1: Add a scale and an orientation to the figure.

*Reply: The scale in Figure 1 cannot be included, as it is a perspective photo, but we have added bar scales to give an idea of the size of the mud volcano. North has been added.*

Figure 2: North and scale are missing.

*Reply: North and scale have been added.*

Figure 3: Correct 'local' in the legend.

*Reply: The legend has been corrected.*

Figure 6: Correct 'Adjustment' and 'Reconstruction'. In Step 3, the white box is missing the word 'Point' in 'Build Dense Cloud'. Also, specify the acronym DEM.

*Reply: The text within the Figure has been corrected.*

Figure 17: in the legends, correct 'Area', sometimes written as 'areae'.

*Reply: Typing errors have been corrected.*

---

## Author Response (AR3)

**Reply to the Comments:**

UAV survey method to Monitoring and analysing geological hazards: The Case study of mud volcano of Villaggio Santa Barbara, Caltanissetta (Sicily).

*Fabio Brighenti, Francesco Carnemolla, Danilo Messina, and Giorgio De Guidi*

lines 11-12 please reword - this does not currently make sense.

> *Reply: The sentence was reworded.*

line 13 "of natural hazards"

> *Reply: We deleted the article.*

lines 13-15 change "The use of Unmanned Aerial Vehicles (UAVs) in contexts of natural hazard presents three main steps for risk assessment and monitoring: pre-post event data acquisition, emergency support and monitoring." to "The use of Unmanned Aerial Vehicles (UAVs) in relation to observation of natural hazards, encompasses three main stages: pre-post event data acquisition, monitoring and risk assessment."

> *Reply:* The sentence has been reworded according to your suggestions

line 16 "to infrastructure within a range ..."

> *Reply: We changed this sentence according to your suggestions*

line 17 change "clues of" to "precursors to" also change "are the uplift and the development of structural features with dimensions ranged from centimetre to decimetre." to "are uplift and the development of structural features with dimensions ranging from centimetres to decimetres."

> *Reply: We changed this sentence according to your suggestions*

line 18 change to "Here we present a methodology for monitoring deformation processes that may be precursory to paroxysmal events at the Santa Barbara mud volcano. This methodology is based on i) the data collection, ii) the Structure from Motion (SfM) processing chain and iii) the M3C2-PM algorithm for the comparison between point clouds and uncertainty analysis with a statistical approach. The objective of this methodology is to detect precursory activity by monitoring deformation processes with centimetre scale precision and a temporal frequency of 1 - 2 months."

> *Reply: We changed these sentences according to your suggestions*

line 25 - "has allowed the generation of"

> *Reply: We changed these sentences according to your suggestions*

line 38 - for 3D reconstruction

*Reply: We deleted the word*

line 43 - change "is spreading" to "is becoming more widely used"

*Reply: The sentence has been reworded according to suggestions*

line 46 - important to study catastrophic natural events

*Reply: The word has been deleted*

line 46 - remove subsidence as an example - this alone is not a catastrophic event

*Reply: We are agree with you and we deleted the word "subsidence"*

line 48 - change allow to acquire to "enable the acquisition of"

*Reply: We modified the sentece according to your suggestion.*

line 78 - change to "At several mud volcanoes (e.g. Ayaz-Akhtarma and Khara Zira Island mud volcanoes in Indonesia), certain geomorphic/structural features have been observed within the year preceding a paroxysmal event (Antonelli et al., 2014; Madonia et al., 2011)."

*Reply: We modified the sentece according to your suggestion.*

line 95 - reword - this does not make sense – do you mean they extend outside the mapped limits of the volcano?

*Reply: we mean that the fractures and the shear lineaments were observed outside the volcano area. We mean as volcano area the area covered by the mud erupted during the eruption of 2008.*

line 97 - in 2002 and 2008

*Reply: we changed the word.*

line 140 - reduced to approximately 4 cm

*Reply: we changed the sentece according to your suggestion.*

In section 2.4 remove unnecessary paragraph indents and reword the repeated use of "After that, after this etc" - for example to following this ...

*Reply: the section 2.4 was rewritten in order to make this section more fluently.*

line 230 - The next step ...

    *Reply: We added the article*

line 234 - reword "the best is ..." this is not very scientific.

    *Reply: we reword the sentence.*

line 243 "Once the ..." and "in order to determine the changes between them"

    *Reply: We modified the sentence according to your suggestion*

line 315 - is to detect significant deformation ...

    *Reply: We changed the word.*

line 316 - this deformation

    *Reply: we corrected the sentece.*

lines 324-325 – reword

    *Reply: the sentence was reworded*

line 360 - The first is ...

    *Reply: we deleted the word.*

line 383 - reword first sentence

    *Reply: the sentence was deleted because is a repetition after is better explained.*

line 386 - small area ... or in a larger region ...

    *Reply: We followed the first suggestion. For the second we decided to write "larger portion" because we are considering not a region but a portion of the mud volcano area.*

line 391 - This monitoring technique is a useful tool ...

    *Reply: We changed the word*

line 395 - uplift can occur ...

    *Reply: We changed the word.*

line 397 - In hazard management ...

    *Reply: We changed the word.*

line 403 - development of …

      *Reply: We changed the word.*

line 416 - and ii) …

      *Reply: We added the conjunction.*

---

## Author Response (AR4)

**Reply to Reviewer 1**

UAV survey method to Monitoring and analyzing geological hazards: The Case study of mud volcano of Villaggio Santa Barbara, Caltanissetta (Sicily)

Fabio Brighenti et al.

A pdf file of the reviewed manuscript is attached as supplement file. The changes in the text are marked with red for rephrasing, reorganized and the changes suggested by the referees.

**Specific comments:**

Please just reword the first sentence: All active geological processes generally determine effects on the ground surfaces such as uplift, subsidence and shear lineaments.

I would suggest "Active geological processes often generate a ground surface response such as uplift, subsidence and faulting/fracturing."

> **Reply**: Thanks for the suggestion the *sentence has been changed.*